# HIV-1 vif mediates ubiquitination of the proximal protomer in the APOBEC3H dimer to induce degradation

Katarzyna A. Skorupka [1,9], Kazuhiro Matsuoka[2,9], Bakar Hassan [3], Rodolfo Ghirlando[4], Vanivilasini Balachandran[1], Ting-Hua Chen [5], Kylie J. Walters [3], Celia A. Schiffer [6], Matthias Wolf [5,7], Yasumasa Iwatani [2,8] ✉ & Hiroshi Matsuo [1] ✉

The APOBEC3 family of cytidine deaminases restricts retroviruses like HIV-1 by mutating viral DNA. HIV-1 evades this restriction by producing Vif, which recruits the Cullin-5 (CUL5) E3 ubiquitin ligase complex to promote APOBEC3 degradation. Here we resolve key aspects of this counter-defense mechanism by determining a 3.6 Å cryo-EM structure of chimpanzee APOBEC3H (cpzA3H) in complex with HIV-1 Vif and three components of the CUL5 E3 ligase-CBFβ, EloB, and EloC (VCBC). The structure captures cpzA3H as an RNA-mediated dimer within the cpzA3H-VCBC complex, allowing us to examine the role of dimerization. We find that ubiquitination occurs specifically at two lysine residues on the Vif-proximal protomer, while the distal protomer remains unmodified. The structural model of the active cpzA3H−Vif−CUL5 E3 ligase holoenzyme reveals spatial preferences for ubiquitin transfer to the targeted lysine residues. These findings enhance our understanding of A3H degradation and suggest new antiviral strategies targeting this host-virus interface.

In recent years, the interplay between host restriction factors and viral evasion strategies has emerged as a critical battleground in the ongoing arms race between viruses and their hosts. Among the diverse array of host restriction factors, apolipoprotein B mRNA editing enzyme, catalytic polypeptide-like 3 (APOBEC3) proteins have garnered significant attention for their potent antiviral activity against retroviruses, including human immunodeficiency virus type-1 (HIV-1)[1–7]. APOBEC3 (A3) proteins are evolutionarily conserved cytidine deaminases that exert robust inhibitory effects on retroviral replication by inducing mutations in the viral genome[4,7–9]. Their cytidine deamination activity causes extensive hypermutation of negative-strand DNA of a newly reverse-transcribed viral genome, which leads to G-to-A mutations within plus-strand DNA and impairs viral replication. The A3-mediated antiviral activity is counteracted by the HIV-1 accessory protein Virion infectivity factor (Vif). Vif impedes A3 activity by hijacking a Cullin-5 (CUL5) E3 ubiquitin ligase complex that contains cellular T-cell transcription cofactor core-binding factor beta (CBFβ), CUL5, Elongin B (EloB), Elongin C (EloC), and RING-box protein 2 (RBX2)[1–7,10–16]. This host-pathogen interaction leads to the formation of polyubiquitin chains on APOBEC3 proteins and their

[1]Cancer Innovation Laboratory, Frederick National Laboratory for Cancer Research, Frederick, Maryland 21702, USA. [2]Clinical Research Center, National Hospital Organization Nagoya Medical Center, Nagoya, Aichi, Japan. [3]Protein Processing Section, Center for Structural Biology, Center for Cancer Research, National Cancer Institute, National Institutes of Health, Frederick, MD, USA. [4]Laboratory of Molecular Biology, National Institute of Diabetes and Digestive and Kidney Diseases, National Institutes of Health, DHHS, Bethesda, Maryland 20892-0540, USA. [5]Molecular Cryo-Electron Microscopy Unit, Okinawa Institute of Science and Technology Graduate University, Okinawa, Japan. [6]Department of Biochemistry and Molecular Biotechnology, University of Massachusetts Chan Medical School, Worcester, MA 01605, USA. [7]Institute of Biological Chemistry, Academia Sinica, Nankang, Taipei, Taiwan. [8]Department of AIDS Research, Division of Basic Medicine, Nagoya University Graduate School of Medicine, Nagoya, Aichi, Japan. [9]These authors contributed equally: Katarzyna A. Skorupka, Kazuhiro Matsuoka. ✉e-mail: yasumasa.iwatani@nnh.go.jp; hiroshi.matsuo@nih.gov

subsequent proteasomal degradation. It represents a crucial mechanism employed by HIV-1 to evade host immune defenses and to establish productive infection in target cells.

Structural studies of APOBEC3 enzymes and Vif have elucidated the binding interfaces between these proteins, revealing key residues and structural motifs critical for protein-protein recognition and complex formation. Recent cryo-electron microscopy (cryo-EM) studies have provided detailed structural insights into the formation of the A3F-Vif, A3G-Vif, and A3H-Vif complexes[17–21], revealing the multifaceted nature of Vif interactions with A3 proteins, where Vif employs distinct interaction interfaces for each A3 protein[22]. Furthermore, more recent cryo-EM studies have also included components of the CUL5 E3 ubiquitin ligase (CUL5, EloB, and EloC), thus providing an initial framework for understanding the process of Vif-mediated APOBEC3 ubiquitination. However, we currently lack complete information about APOBEC3H (A3H)[11,23–30], which is unique among the APOBEC3 family members in that it forms a stable homodimer mediated by double stranded RNA (dsRNA), which bridges the protomer–protomer interface[31–34], and was shown to be critical for A3H activity. Despite the functional relevance, the reported A3H-Vif-CBFβ-EloB-EloC-CUL5 (A3H-VCBC-CUL5) complex structure resolved only one A3H protomer[21], prompting questions about whether the missing protomer has a distinct role in Vif interaction and the ubiquitination mechanism.

Previously, we and others reported that chimpanzee APOBEC3H (cpzA3H) is degraded by HIV-1 Vif more efficiently than human APOBEC3H derived from haplotype II[14,35], and that the cpzA3H protein exhibits relatively higher solubility[33]. In this study, we determined a cryo-EM structure of the chimpanzee A3H (cpzA3H) in complex with Vif of HIV-1 NL4-3 strain, CBFβ, EloB, EloC (VCBC) at 3.6 Å resolution. Our cryo-EM structure allowed us to map interactions between cpzA3H and Vif, Vif and Vif, and Vif and RNA, and to identify structural determinants involved in the ubiquitination of cpzA3H. We complemented our cryo-EM studies with biochemical and mutational analysis, as well as molecular modeling and molecular dynamics (MD), to demonstrate that ubiquitination specifically targets two lysine residues on the cpzA3H protomer that directly interacts with Vif, while the distal protomer remains unmodified. Nonetheless, this targeting is sufficient for A3H degradation via the proteasome. Furthermore, we propose that dsRNA might play a more central regulatory role in directing A3H towards Vif-mediated degradation.

## Results

### Architecture of the A3H-VCBC Complex

In this study, we used in vitro reconstituted cpzA3H-bound Vif/core binding factor subunit β (CBFβ)/Elongin B (EloB)/Elongin C (EloC) complex (hereafter referred to as the cpzA3H-VCBC complex) that also included dsRNA (previously reported to facilitate A3H dimerization). The peak corresponding to the cpzA3H-VCBC complex was separated from other fractions using size-exclusion chromatography (Supplementary Fig. 1), and its size was verified to be the expected molecular weight by analytical ultracentrifugation sedimentation (Supplementary Fig. 2). We observed that the cpzA3H-VCBC complex exists in an equilibrium between monomeric, dimeric, and, to a lesser extent, higher multimeric assemblies at our working protein concentration of 30–40 μM. This dissociation between cpzA3H and VCBC appeared to be rapid, as significant amounts of free cpzA3H and free VCBC were observed in the equimolar mixture of cpzA3H and VCBC (Supplementary Fig. 2c). Therefore, to stabilize the complex for cryo-EM analysis, we used chemical crosslinking via Bis(sulfosuccinimidyl)suberate (BS3) to allow capture of the dimeric assemblies of the cpzA3H-VCBC complex. We designated each cpzA3H-VCBC complex as a "protomer" and solved two structural models—one without imposed symmetry (C1) and another with C2 rotational symmetry.

As illustrated in Fig. 1, each model included density for A3H, the dsRNA that mediates A3H dimerization, and all the components of the VCBC complex organized into a dimer. When we built a model without imposing C2 rotational symmetry, we observed that each of the subunits had regions of low-resolution density, which affected our ability to model in all the components of this assembly. Overall, in one of the protomers (Fig. 1, black), the dimerization of cpzA3H via dsRNA was clearly visible, although the density for the distal A3H molecule was insufficient for de novo atomic modeling. The Vif-CBFβ part of VCBC was well-resolved, but we observed a weaker density for EloB and EloC. On the other hand, in the second protomer (Fig. 1, pink), we observed a strong density for the VCBC component, including EloB and EloC. However, the density of the distal A3H molecule was weaker. This weak density is likely due to preferred orientation of the complex in the ice layer (Supplementary Fig. 3 and 4) and/or potential damage incurred during the orientation process at the water-air interface. Nevertheless, our final reconstruction of the complex without imposed symmetry was around 4 Å.

To improve the resolution and given that the two cpzA3H-VCBC units in the assembly demonstrated a rotational symmetry in the C1 structural model, we applied C2 symmetry, resulting in a reconstruction with a global resolution of 3.6 Å (Supplementary Figs. 3 and 4). Further post-processing using the EMready neural network algorithm[36] enabled more precise placement of amino acid side chains within the EM-map (Supplementary Fig. 5). The C2 model displayed well-defined units composed of Vif, CBFβ, cpzA3H, and dsRNA. Due to the asymmetric distribution of proteins in the C1 structure, the EM-map corresponding to EloC was weaker, while the densities of peripheral molecules, including EloB and the distal A3H molecule, were almost averaged out in the C2 model. Within a single protomer, the Vif and CBFβ structure closely resembles the crystal structure of the VCBC-CUL5 complex (PDB ID: 4n9f)[37]. The cpzA3H molecule and dsRNA align well with the previously published crystal structure of the cpzA3H dimer (PDB ID: 5z98)[33]. The local resolution of the map varied from 3.0 to 7.0 Å, with the highest resolution recorded at the core of the complex where cpzA3H molecules interact with Vif, and the lowest at the peripheral regions like EloB and distal A3H (Supplementary Fig. 4). Taken together, we solved a cryo-EM structure of the A3H-VCBC complex that revealed the presence of the A3H dimer with one A3H protomer engaging Vif (proximal), and the other protomer (held together "glued" via dsRNA) positioned distal from Vif and the components of the ubiquitination machinery. Therefore, our complex represents a structural framework for rationalizing molecular determinants of Vif-mediated A3H ubiquitination.

### Interactions between cpzA3H and Vif

Within each cpzA3H-VCBC assembly, we observed extensive interactions between the proximal cpzA3H (cpzA3H$^p$) and Vif, with a buried area of 692 Å$^2$. These interactions primarily involve the α3 and α4 helices of cpzA3H$^p$ and the β-sheet (β2-β6 strands) of Vif, consisting of both hydrophobic and electrostatic contacts. Many of these interactions involve amino acids located in the α3 helix of A3H (Fig. 2a). The amino acid at position 86 is a key determinant of whether human A3H is resistant (G, A, D, K86 or sensitive (S86)) to degradation by HIV-1 Vif[12,24,26,38]. This importance suggests that the hydroxymethyl group of serine may be directly involved in binding Vif[14,15]. Indeed, the cpzA3H$^p$-Vif interface shows that the hydroxyl group forms a hydrogen bond with the carboxyl group of E45 on Vif (Fig. 2b). E45 is a conserved residue across HIV-1 strains, highlighting the significance of this interaction[39]. The cpzA3H$^p$-Vif interface also involves hydrophobic interactions mediated by W90 of A3H$^p$ and the aromatic ring of Vif H56, along with Van der Waals interactions involving the side chains of R41, F39, and E45 of Vif (Fig. 2b). Additionally, L125 contributes to a hydrophobic core formed by W90 and L125 of cpzA3H$^p$, and F39 of Vif,

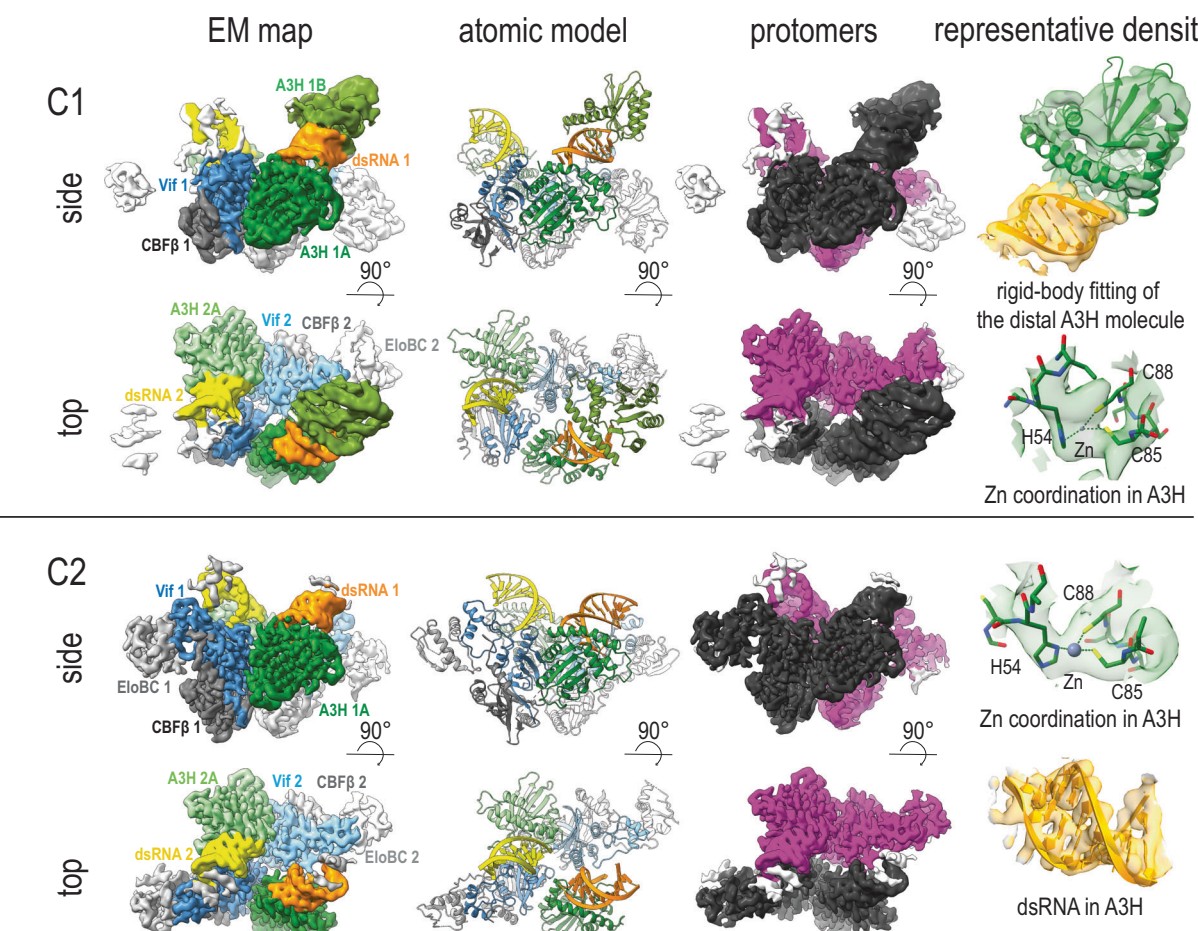

**Fig. 1 | Cryo-EM structure of the cpzA3H-VCBC complex. Top panel**: 4 Å cryo-EM reconstruction (EM map), the atomic model, and arrangement of protomers (pink and black) of the cpzA3H-VCBC complex without symmetry applied (C1). Two molecules of cpzA3H (dark green and green) complexed with dsRNA (orange), Vif (dark blue), and CBFβ (dark grey) form the black protomer. The pink protomer consists of one molecule of cpzA3H (light green) bound to dsRNA (yellow), Vif (light blue), CBFβ (light grey), and EloBC (white). Representative cryo-EM densities of distal A3H, RNA, and the Zn-coordinating region of proximal A3H are shown in semi-transparent densities superposed with corresponding atomic models.

**Bottom panel**: 3.6 Å cryo-EM reconstruction, the atomic model, and arrangement of protomers of the cpzA3H-VCBC complex with C2 symmetry applied. Representative cryo-EM densities of RNA and the Zn-coordinating region of proximal A3H are shown in semi-transparent densities superposed with corresponding atomic models. Both black and pink protomers contain identical component molecules: one molecule of cpzA3H (dark green in the black protomer, light green in the pink protomer), dsRNA (orange and yellow, respectively), Vif (dark blue and light blue, respectively), and CBFβ (dark grey and light grey, respectively).

which is in agreement with previous findings that L125E or L125K substitutions partially confer resistance to Vif[14].

Additionally, several polar/electrostatic interactions also play a significant role in stabilizing this extensive interface, including the carboxyl group of cpzA3H[P] D94 that forms an electrostatic interaction with the guanidine group of R41 in the β2 strand of Vif. This is in agreement with a previous observation that the R41A substitution disrupts the degradation of A3H[15]. The nearby cpzA3H Q97, a critical determinant of resistance to Vif-induced degradation, is positioned at the C-terminal end of the α3 helix and forms a hydrogen bond with the sidechain amino group of K63 on the β4 strand of Vif. This hydrogen bond increases cpzA3H's sensitivity to Vif and helps explain why human A3H, which has lysine at position 97, is more resistant to Vif-induced degradation. R93, located in the short loop between β5 and β6 of Vif, is stabilized by electrostatic interactions between its guanidinium group and the acidic side chains of D100 and E70 of cpzA3H[P]. On the α4 helix of cpzA3H[P], the side chains of E121 and R124 form hydrogen bonds with Y30 sidechain and K36 mainchain of Vif, respectively, while L125 engages in a hydrophobic interaction with F39 of Vif.

A total of ten residues from cpzA3H[P] and nine residues from Vif contribute to the formation of the A3H[P]-Vif interface (Fig. 2c). The Vif-

cpzA3H[P] interactions observed in our structure are consistent with our previous mutational studies, where we systematically substituted surface residues of cpzA3H and tested their effects on Vif-induced degradation[14]. To further assess the biological relevance of these interactions, we performed additional mutational experiments targeting specific Vif residues involved in the cpzA3H[P]-Vif interactions. These experiments showed that mutations such as F39A, R41A, H56A, K63A, K92A, or R93A rescued cpzA3H from Vif-induced degradation, with F39A, R41A and H56A, and K63A having the most significant effects (Fig. 2d). Interestingly, H48 of Vif, a residue implicated as important for Vif's ability to degrade A3H[11], does not seem to be part of the Vif-cpzA3H[P] interactions (Fig. 2b, c).

In addition to these interactions, the proximal cpzA3H also interacts with the Vif molecule from the other protomer, with buried area of 245 Å[2]. This interaction is mediated by the lysine-rich loop 3 of cpzA3H and the tip of the helix α1/β5-β6 loop of Vif (Fig. 3a, b). The protein-protein interactions between these two molecules may be limited, as they are positioned relatively far apart, with a distance of approximately 6.5 Å between the backbone of loop 3 of cpzA3H and α1 of Vif. A key interaction occurs between K52 of cpzA3H, which was implicated as important for the ubiquitination of human A3H[21], and the

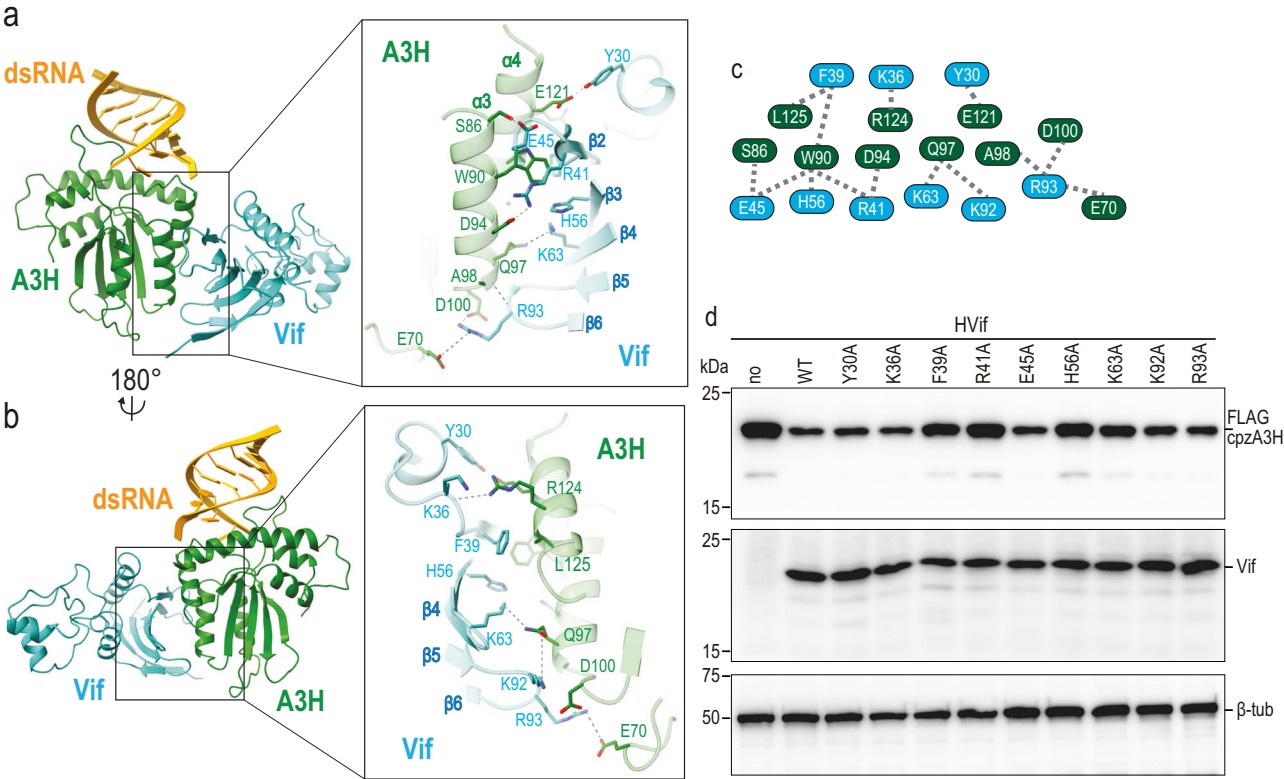

**Fig. 2 | Molecular details of the cpzA3H-Vif interface. a, b** Atomic model of cpzA3H (green), dsRNA (yellow), Vif (cyan), with close-up views of the interface between cpzA3H and Vif (middle panels). Potential hydrogen bonds are indicated by dashed lines. Part of the atomic model was hidden to show the interacting amino acids clearly. **c** Schematic of the amino acid residues involved in this cpzA3H-Vif interactions. **d** Effect of amino acid substitution at the cpzA3H-Vif interface. The individual Vif amino acid residues highlighted in (**c**) were substituted with alanines, and the degradation of FLAG-tagged cpzA3H (top) was assessed. The intracellular levels of cpzA3H and Vif were probed 40 h post-transfection by Western blotting using anti-FLAG and anti-Vif mAbs. Blotting for ß-tubulin with polyclonal Abs (ß-Tub) was used as a loading control. The cpzA3H stability in the absence of Vif (no, lane 1) and presence of wild-type Vif (WT, lane 2) were used as controls. The results are representative from three independent experiments.

main chain carboxyl groups of H80 and L81 of Vif (Fig. 3b). Additionally, the guanidino group of Vif R15 appears to play a supportive role in stabilizing the interactions by providing an electrostatic interaction to H80 (Fig. 3b).

Our structure of each cpzA3H–VCBC protomer closely resembles the recently published hA3H–VCBC structure by Ito, Chen, and colleagues[21], with an RMSD of less than 1 Å for each component protein (Supplementary Fig. 6). Key A3H residues involved in Vif binding are conserved between the two structures, however, our structure reveals A3H–Vif interactions across protomers, as shown in Supplementary Fig. 6c.

Taken together, our cryo-EM structure serves to rationalize a substantial volume of biochemical and mutational studies, including the one we performed here. Overall, the interface between the proximal A3H and Vif is complex and mediated by both hydrophobic and polar interactions. In terms of inter-assembly interactions, given that the interface involves one of the lysines (K52) previously reported to be important for ubiquitination, we hypothesize that this interface is functionally relevant, as it structurally configures the ubiquitination-target lysines (K50 and K51; see also "Ubiquitination mechanism of cpzA3H" section for more on the role K50, K51, and K52). However, future experiments are needed to determine whether and/or the abundance of the cpzA3H-VCBC dimer in cells.

### Double-stranded RNA mediates intra- and inter-protomer interactions

We and others have shown that A3H forms dimers with double-stranded RNA (dsRNA)[32]. Our cryo-EM map supports this finding, revealing that dsRNA binds to A3H (Figs. 1 and 3a). Due to the limited resolution of our cryo-EM data, we could not determine the exact RNA sequence in our model. Instead, we used the RNA sequence identified in the cpzA3H crystal structure that we previously reported, as the A3H protein was prepared using the same method for crystallography and cryo-EM. From this modeling, we noted that cpzA3H interactions with dsRNA were similar to what was previously described[33]. For example, as observed previously, dsRNA serves as a molecular bridge that simultaneously binds to two molecules of A3H, forming a ternary complex shaped like a dumbbell. These two molecules engage dsRNA using two basic surfaces, creating a positively charged channel that accommodates the negatively charged RNA strand (Supplementary Fig. 7). Although the density for the "distal" A3H, the A3H unit positioned away from Vif, was of lower resolution and challenging to model, we have nevertheless taken advantage of the available crystal structure to build A3H[D] into the low-resolution density. As mentioned, the observed poses and interactions did not display large deviations from how the A3H dimer engaged dsRNA in isolation.

In addition to A3H-dsRNA interactions, our structure revealed an interaction between the cpzA3H-bound RNA and the Vif molecule from the other protomer (buried area of 195 Å²) in the dimeric cpzA3H-VCBC assembly (Fig. 3a, c). Multiple positively charged residues, including R23, H27, K157, and K160 of Vif are closely positioned to the negatively charged phosphate backbone of the RNA (Fig. 3c, d). To investigate whether these interactions are functionally relevant, we tested Vif variants R23A, H27A, K157A, K160A, P161A, R15A, H80A, and L81A for their ability to degrade cpzA3H in HEK293T cells (Fig. 3e). We observed that R15A, L81A, K160A, P161A, and to a lesser extent, H27A Vif

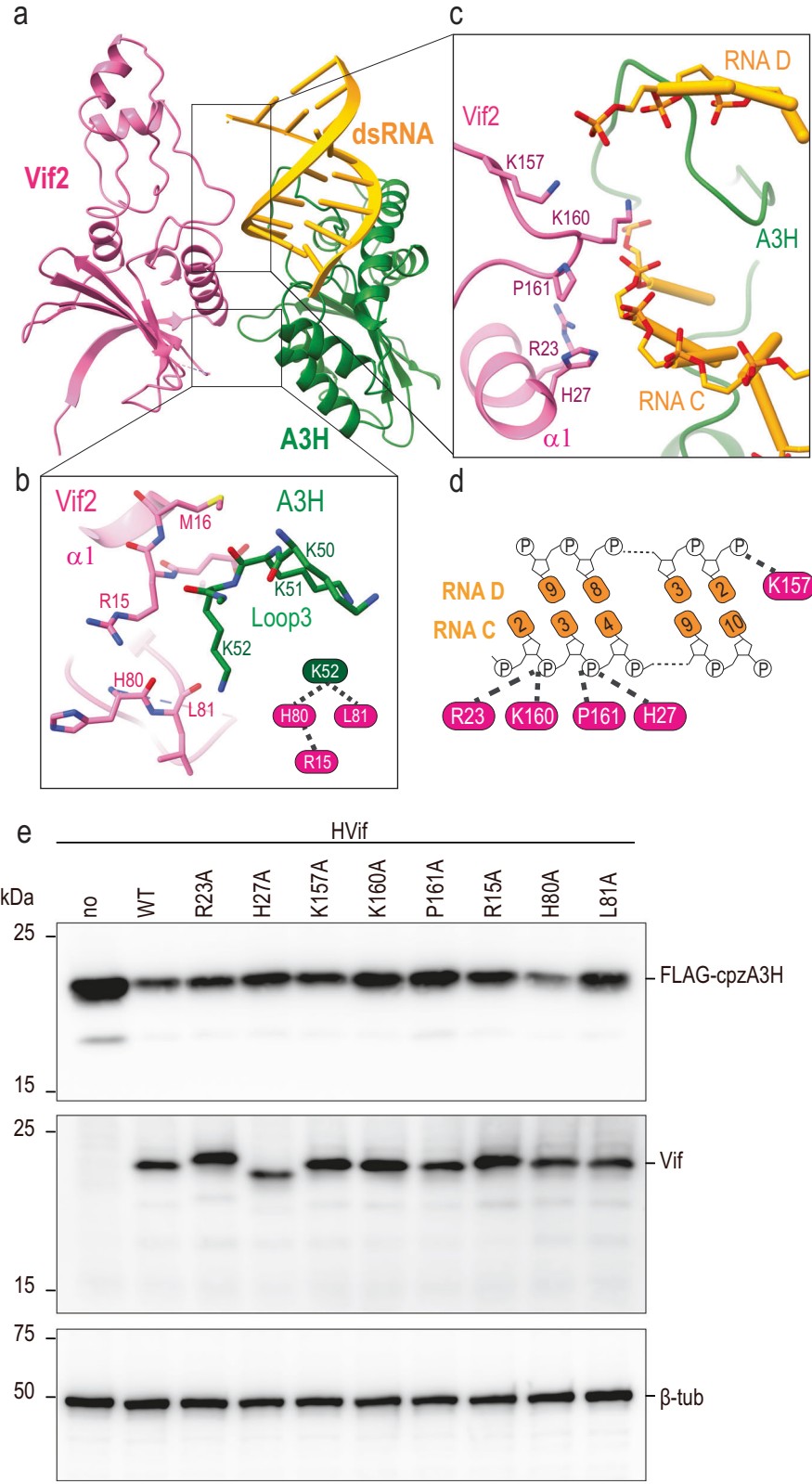

**Fig. 3 | Across protomers RNA-Vif and cpzA3H-Vif interaction. a** Atomic model of the cpzA3H-bound dsRNA from protomer 1 (green and yellow, respectively) and Vif from protomer 2 (pink). **b** Close-up view of the across-protomers interface between cpzA3H and Vif. The amino acids involved in these interfaces are represented in stick model format. **c** Close-up view of the interface between RNA and Vif amino acid residues across the protomers. **d** Schematic of the Vif amino acid residues involved in the RNA interactions across protomers. **e** Effect of amino acid substitutions at these RNA-Vif and cpzA3H-Vif interfaces on cpzA3H degradation. Individual Vif amino acid residues highlighted in (**b**) and (**d**) were substituted with alanines, and the degradation of FLAG-tagged cpzA3H was assessed (top). The intracellular levels of cpzA3H and Vif were probed 40 h post-transfection by Western blotting using anti-FLAG and anti-Vif mAbs. β-Tub was used as a loading control. The cpzA3H stability in the absence of Vif ("no", lane 1) and presence of wild-type Vif ("WT", lane 2) were used as controls. The results are representative from three independent experiments.

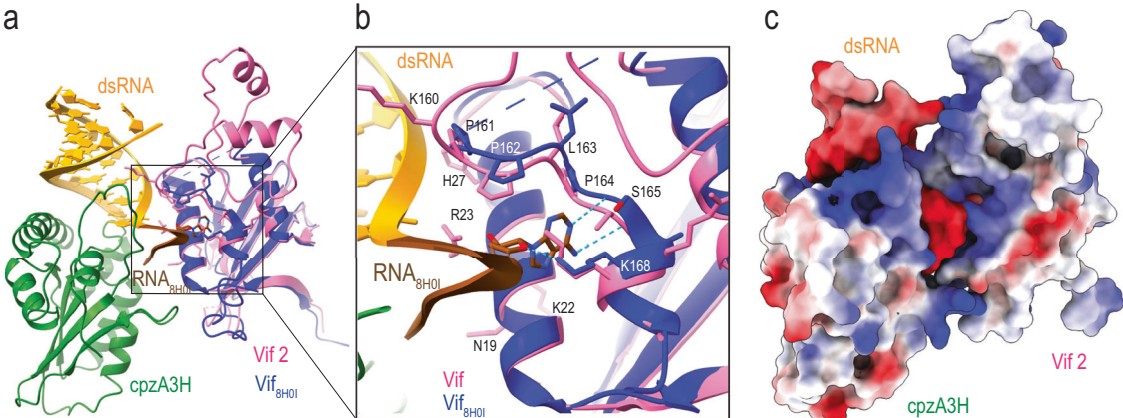

**Fig. 4 | Interaction between RNA and Vif. a** Superimposed structures of the cpzA3H-VCBC complex (with cpzA3H in green, RNA in yellow and Vif across protomers in pink) and the Vif (blue) and RNA (brown) molecules from the A3G-VCBC complex (PDB ID: 8H0I)[20]. The RMSD between the two Vif molecules is 0.62 Å. In the cpzA3H-VCBC structure, nucleotides 16–19 from 8H0I's RNA strand were connected to chain G of cpzA3H-bound dsRNA (this study). **b** The panel expands on the Vif 161-PPLPS-165 region, showing an adenine molecule (from A3G-VCBC structure) with hydrogen bonds to P164, S165, and K168 (cyan dotted lines). **c** Electrostatic potential of cpzA3H, cpzA3H-bound dsRNA and Vif, including extended RNA strand from (PDB ID: 8H0I)[20] as shown in panel (**a**). RNA-interacting residues of cpzA3H and Vif form a continuous positively charged surface accommodating the negatively charged RNA. The surface was colored in ChimeraX[63] according to the electrostatic potential, ranging from -10.0 kT/e (red) to +10.0 kT/e (blue).

mutations decreased the ability of Vif to degrade cpzA3H to varying degrees., while the H80A, R23A and K157A mutations had no apparent effect on cpzA3H degradation in cells (Fig. 3e). It is noteworthy that R23 and H27 residues are located on the α1 helix of Vif in proximity to N19, K22, and K26 which interact with the phosphate backbone of the RNA bound to A3G, as demonstrated by A3G-Vif complex structures[18–20]. This suggests that the α1 helix of Vif may function as an RNA-binding domain, facilitating interactions with RNA-bound proteins, including other A3s. In addition, the Vif-RNA interactions we identified here are similar to the interaction observed in the A3G-Vif-CBFβ complex structure, where Vif residues P164, S165, and K168 interact with RNA[20] (see superimposed A3G-Vif-CBFβ complex structure (PDB: 8H0I, blue) and our Vif structure (pink); Fig. 4a). The electrostatic potential map (Fig. 4b, c) illustrates that positively charged amino acids of both cpzA3H and Vif form a continuous surface across the protomer interface, where an RNA strand extends to interact with the Vif 161-PPLPS-165 motif. These results suggest that Vif-dsRNA binding interactions play a role in maintaining the dimeric cpzA3H-VCBC assembly, which seems to be important for effective ubiquitination and proteasomal degradation. Additionally, given that similar interactions were documented in the APOBEC3G (A3G)-Vif structures[18–20], and shown to be essential for Vif-induced A3G degradation[20], these results suggest that the presence of RNA might be a critical feature that directs A3s towards Vif-dependent proteasomal degradation.

## Vif-Vif Interface

The structure of the dimeric cpzA3H-VCBC assembly is symmetric, with the center of symmetry located between the two Vif molecules. This Vif-Vif (called Vif-1-Vif-2 hereafter) interface is mediated by a relatively small surface area (buried area of 222 Å²) composed of residues in the β2-β3 and β4-β5 loops. Key amino acids at this interface include H42, H43, Y44, P49, W70, and H80 (Fig. 5a–c), which interact primarily through their side chains. Intriguingly, most of these residues (residing in the Vif 40-YRHHY-44 region) were previously shown to affect the degradation of human A3H[15]. While R41 plays a critical role at the cpzA3H-Vif interface (Fig. 2b), the other residues in this region do not directly interact with cpzA3H, raising a question about their contribution to the ubiquitination process. To investigate further their importance, we tested Vif variants H42A, H43A, Y44A, P49A, W70A, and H80A for their ability to degrade cpzA3H in HEK293T cells. Our

results indicate that the Y44A and P49A mutations, and to a lesser extent H42A and W70A, reduce cpzA3H degradation, while the H43A and H80A mutations have no apparent effect on cpzA3H levels in cells (Fig. 5d).

One possible way residues mediating Vif1-Vif2 interactions may impact ubiquitination efficacy is by affecting the stability of the dimeric cpzA3H-VCBC complex. To test this, we performed a 1000 ns molecular dynamics (MD) simulation. The results show that the WT cpzA3H-VCBC dimer retains its structural integrity throughout the simulation and remains largely unchanged from the initial structure (Supplementary Fig. 8a–c). However, introducing the H43A/W70A/H80A triple mutations in Vif affected the dimer's stability, causing one protomer to rotate away from the other (Supplementary Fig. 8d–g). This rotation indicates a destabilization and, in turn, may introduce sufficient perturbation to the assembly to prevent ubiquitination of A3H from taking place. Based on insights from our cryo-EM structure and MD simulation, we generated a Vif triple mutant (H43A/W70A/H80A), targeting residues located at the Vif–Vif interface. We then performed analytical ultracentrifugation (AUC) using a non-cross-linked VCBC sample. Our results show that these mutations significantly reduced the presence of dimeric and higher-order oligomeric forms of VCBC (Supplementary Fig. 2e and f). Altogether, these data imply a model whereby Vif1-Vif2 interactions provide support for stable cpzA3H-Vif CUL5 E3 ligase formation, thereby facilitating efficient A3H ubiquitination; future experiments are needed to test this model however.

## Ubiquitination mechanism of cpzA3H

Identification of ubiquitination sites may provide insights into the molecular mechanism of the ubiquitination step of cpzA3H. Our structure includes EloB and EloC, essential components of CUL5 E3 ubiquitin ligase, and therefore offers a suitable model to examine some of the issues related to this question. For example, given that only one protomer from the cpzA3H dimer forms interactions with Vif (cpzA3H^P), while the distal cpzA3H (cpzA3H^D) does not seem to be involved, we were interested in exploring which cpzA3H molecule is the primary site of ubiquitination, and whether the other plays a scaffolding role that ensures successful ubiquitination.

We constructed an in vitro ubiquitination system for cpzA3H using HIV-1 NL4-3 Vif and CUL 5 E3 ligase complex[20]. A3H, the VCBC complex, CUL5/RBX2 complex, ARIH2 and UbcH7 were separately

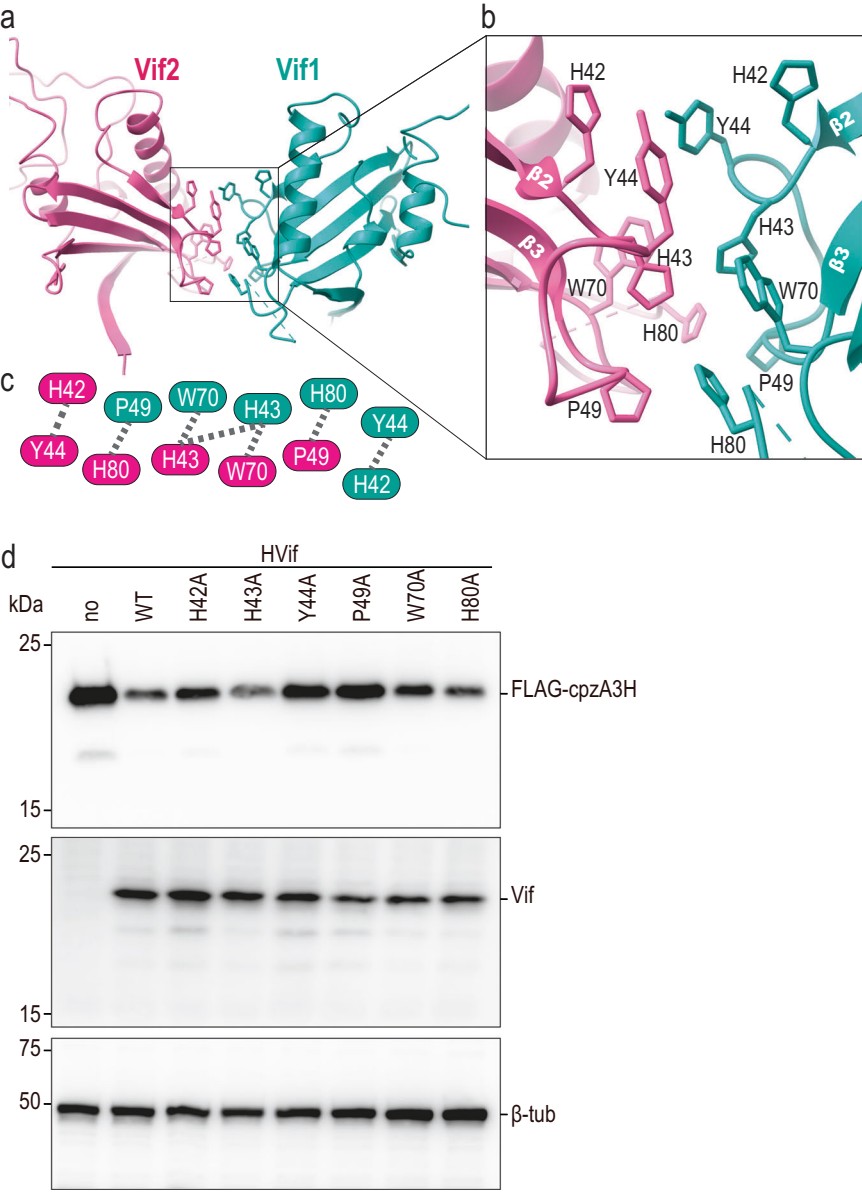

**Fig. 5 | Across protomers Vif-Vif interface. a** Atomic model of the Vif-Vif interface formed between the protomers of the cpzA3H-VCBC complex. **b** Close-up view of the interface between Vif 1 (cyan) and Vif 2 (pink). The amino acids involved in this interface are represented in stick model format. **c** Schematic of the amino acid residues involved in Vif-Vif interactions. **d** Effect of amino acid substitutions at the Vif-Vif interface on cpzA3H degradation. The individual Vif amino acid residues highlighted in (**c**) were substituted with alanines, and the degradation of FLAG-tagged cpzA3H was assessed (top). The intracellular levels of cpzA3H and Vif were probed 40 h post-transfection by Western blotting using anti-FLAG and anti-Vif mAbs. An anti-ß tubulin polyclonal Abs (β-Tub) was used as a loading control. The cpzA3H stability in the absence of Vif ("no", lane 1) and presence of wild-type Vif ("WT", lane 2) were used as controls. The results are representative from three independent experiments.

purified prior to the in vitro ubiquitination experiment, and ARIH2 was charged with a ubiquitin using UBE1 and UbcH7, and UbcH3 was added for extension of ubiquitin chain (Fig. 6a). Ubiquitinated cpzA3H was found in the in vitro ubiquitination mixture by immunoblotting using an anti-A3H antibody. Mono-ubiquitination of cpzA3H was strongly observed, and the reaction mixture was submitted to mass spectrometry analysis (Fig. 6b). We divided the reaction mixture into two samples and used trypsin or Lys-N enzyme for protein digestion to achieve better amino acid sequence coverage. After the protease digestion, half of the sample was enriched for ubiquitinated peptides, whereas the other half was left without enrichment. Both samples were then analyzed by LC-MS/MS. cpzA3H has 12 lysines and our MS analysis could detect most of lysine-containing peptides except for K117 and K181. Of all 10 detected lysines, ubiquitination was identified on K50,

K51 and K168 (Fig. 6c). Ubiquitination of K27, K153 and K161 was less frequently detected compared with K50 and K51, and ubiquitination of K52 was rarely observed, and only in double ubiquitination with K51 or triple ubiquitination with K50 and K51 (Fig. 6d). K27, K50, K51 and K52 have been previously identified as the ubiquitination sites for human A3H[21], and our data generally support these findings. We mutated seven lysines identified by our mass spectrometry experiments to arginine, including K27R, K50R, K51R, K52R, K153R, K161R and K168R (called cpzA3H-7KR here after), and monitored Vif-induced degradation of cpzA3H in HEK293T cells. Seven K to R mutations protected cpzA3H from degradation significantly (Fig. 6e). We confirmed that K50 and K51 were preferred ubiquitination sites for cpzA3H as reverse mutation of 50 K or 51 K in cpzA3H-7KR rescued Vif sensitivity and achieved degradation of the protein at wildtype level (Fig. 6e).

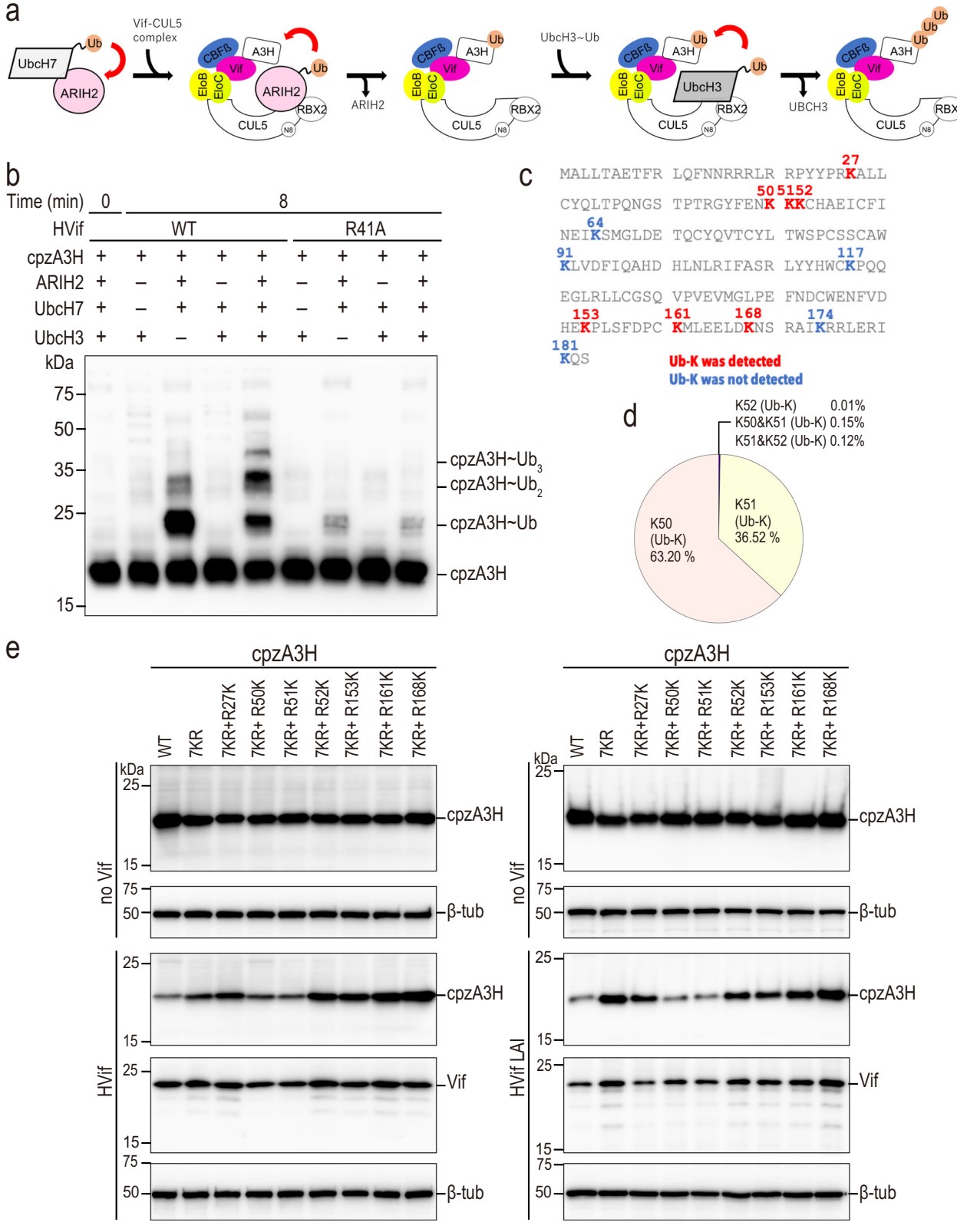

To test whether the distal A3H can be ubiquitinated and degraded, we generated cpzA3H-W90A variant that cannot bind to Vif and co-expressed it with wild-type cpzA3H in HEK293T cells, then monitored their degradation. We confirmed that WT and W90A cpzA3H proteins form heterodimers with comparable efficiency to WT–WT and W90A–W90A homodimers when FLAG- and myc-tagged cpzA3Hs were coexpressed in HEK293T cells (Supplementally Fig. 9). Our

experiments indicated that only wild type cpzA3H was degraded by NL4-3 Vif, NL4-3 N48H Vif, and LAI Vif, suggesting that only the Vif-proximal/binding cpzA3H protomer was ubiquitinated (Fig. 7). Additionally, we confirmed this result by expressing wild-type cpzA3H and the W90A variant as a single transcript containing a P2A site between the two genes. This approach generates an equal number of wild-type and W90A molecules and, importantly, provides an unbiased

**Fig. 6 | K50 and K51 are critical ubiquitination sites of cpzA3H. a** The schematic overview of Vif-induced ubiquitination of A3H. ARIH2 is responsible for transferring the first Ub molecule onto A3H. The UbcH3 then elongates the Ub chain. **b** In vitro ubiquitination of cpzA3H by NL4-3 Vif WT (lanes 2–5) and the R41A variant (lanes 6–9). Ubiquitinated cpzA3H proteins were detected by Western blotting using an anti-A3H rabbit polyclonal Abs. **c** Ubiquitinated lysine (red) of cpzA3H, detected by mass spectrometry, are highlighted in the amino acid sequence of cpzA3H. **d** Comparison of K50, K51, and K52 ubiquitination. The frequency (%) with which peptides containing ubiquitinated K50, K51, K52, and their combinations are

detected in the trypsin-digested K-ε-GG-enriched sample is shown in the pie chart. **e** cpzA3H-7KR contains K27R, K50R, K51R, K52R, K153R, K161R, and K168R mutations and protects cpzA3H from both NL4-3 (left panel) and LAI Vif (right panel) induced degradation (lane 2). Each mutation was reversed within cpzA3H-7KR to generate cpzA3H-7KR-R27K, -R50K, -R51K, -R52K, -R153K, -R161K and -R168K (lanes 3 to 9). The 7KR + R50K (lane 4) and 7KR + R51K (lane 5) variants showed protein degradation efficiency equivalent to wild-type cpzA3H (lane 1). All proteins were expressed without any tag and detected by anti-A3H rabbit polyclonal Abs.

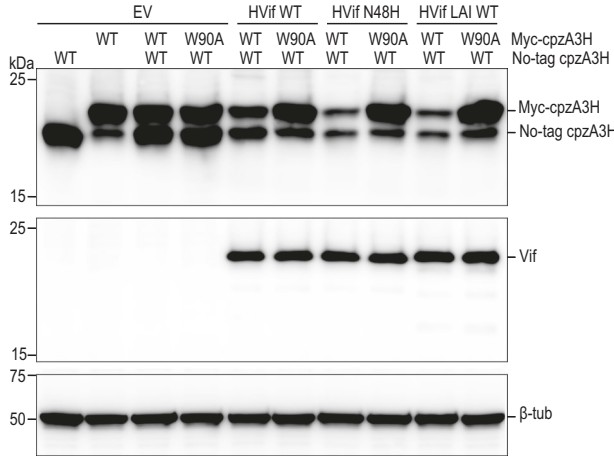

**Fig. 7 | Vif induces degradation of proximal/Vif-Bound cpzA3H molecule.** HEK293T cells were co-transfected with either an empty vector (no Vif) or HIV-1 Vif expression plasmids, alongside the untagged cpzA3H (WT) and myc-cpzA3H (WT or W90A) expression plasmids. Independent vectors were used for each cpzA3H construct. At 40 h post-transfection, the intracellular levels of cpzA3H were analyzed by western blotting using an anti-A3H rabbit polyclonal antibody. Vif expression levels were detected with an anti-Vif mAbs. The ß-Tub was used as a loading control. We tested NL4-3 WT Vif (lanes 5 and 6), the NL4-3 N48H Vif variant (lanes 7 and 8), and LAI Vif (lanes 9 and 10) for their ability to degrade cpzA3H. Both myc-tagged and untagged WT cpzA3H were similarly degraded by all three Vif proteins (lanes 5, 7, and 9). In contrast, degradation occurred only for WT cpzA3H, which binds to Vif, when WT and W90A cpzA3H were co-expressed (lanes 6, 8, and 10).

opportunity for the formation of the wild-type/W90A heterodimer. The results from this P2A experiment fully support the findings obtained using two separate vectors (Supplementary Fig. 10). Taken together, our biochemical and structural data indicate that efficient degradation of cpzA3H by Vif requires that all the interfaces we discussed here are maintained and all the components of the dimeric cpzA3H-VCBC complex are engaged. This ensures productive ubiquitination of the cpzA3H$^P$, on lysines 50 and 51, as they are positioned most closely to the E3 ligase machinery.

### Modeling of the cpzA3H-Vif-CUL5 E ligase holoenzyme complex supports the role of K50 and K51 as the primary ubiquitination sites

To further contextualize the A3H ubiquitination process, we generated a model of the cpzA3H-Vif-CUL5 E3 ligase holoenzyme complex by using our structure and previously determined crystal and cryo-EM structures of the Cullin E3 ligase components (Fig. 8a) (PDB ID: 7B5L, 7B5M, 7ONI, 8FVJ[21,40,41]). Our model depicts the so-called active structure of the Cullin E3 ligase[40], which is primed for ubiquitin transfer to the target protein. The dimeric form of the cpzA3H-VCBC complex can fit into the Vif-CUL5 E3 ligase complex, allowing it to receive a ubiquitin molecule at K50 or K51 on the proximal cpzA3H in one of the protomers. The distances between K50 and K51 of cpzA3H and G76 of

ubiquitin are 12 Å and 13 Å, respectively (Fig. 8b). Other lysine residues, which are ubiquitinated to a much lesser extent than K50 and K51, are located farther from the E3 ligase-bound ubiquitin, supporting the preference for K50 and K51 as the primary targets for ubiquitination. While the cpzA3H-VCBC dimer can fit into the CUL5 E3 ligase complex, only the cpzA3H molecule in one protomer is likely to be efficiently ubiquitinated, as the other protomer is positioned too far from the E3-bound ubiquitin. Therefore, it is plausible that each protomer forms its own E3 ligase complex, which may enhance the efficiency of ubiquitination and degradation of cpzA3H molecules.

## Discussion

Our cryo-EM structure of the cpzA3H-VCBC complex reveals a dimeric assembly consisting of two protomers of the cpzA3H-VCBC complex. Overall, Vif serves as a master scaffold, forming distinct interfaces with A3H, CBFβ, EloB, and EloC within each protomer, as well as with Vif and dsRNA from the other protomer. To summarize Vif's capacity to interact with multiple molecular partners, its binding interfaces are depicted in Supplementary Fig. 11. These interactions stabilize dimerization and suggest that, although the existence of this dimeric form in cells remains to be demonstrated, the dimer seems to represent a more stable form of the cpzA3H-VCBC complex. Our ultracentrifugation sedimentation experiments support these conclusions, since dimers of cpzA3H-VCBC, as well as VCBC are observed. Additionally, A3G-VCBC complex structures also exhibit dimeric assemblies[18–20], although the dimerization is mediated not by the Vif-Vif interface but by A3G-A3G and A3G-Vif interactions. Nonetheless, in both cases, dimerization was considered important for ubiquitination and proteasomal degradation, illustrating a high degree of plasticity in how Vif mediates complex assembly to achieve effective ubiquitination. Recent structural studies of various E3 ligases, including CUL9[42], UBR5[43–45] and BIRC6[46–48], have revealed that these ligases exist in hexameric, tetrameric, and dimeric forms, respectively. Importantly, an increasing number of studies have demonstrated the functional relevance of E3 ligase multimerization in processes such as activation/inactivation, substrate recognition, and the attachment of polyubiquitin chains to target proteins[49,50]. These findings underscore the complexity of E3 ligase multimerization and highlight the importance of understanding how these structures regulate their functional roles. At this point, although dimerization and other types of multimerization, such as tetramerization observed in the A3F-Vif-CBFβ complex[17], appear to be intrinsic features of APOBEC3-Vif complex structures, their functional significance for HIV infection remains elusive and will require further studies.

Beyond insights into the overall architecture of the cpzA3H-VCBC complex, our structure allowed us to rationalize many previous observations, as we documented throughout the results section. In addition to those examples, our structure may also explain the reasons for the distinct neutralization properties of A3H. Whereas the majority of Vif variants from HIV-1 subtypes are able to degrade A3F and A3G proteins, they displayed varied efficacy against A3H that was highly dependent on the presence of F39 and H48 in Vif[11]. The structure of the cpzA3H-VCBC complex now reveals that Vif residue F39 is oriented toward W90 of cpzA3H, serving as a key residue in forming the

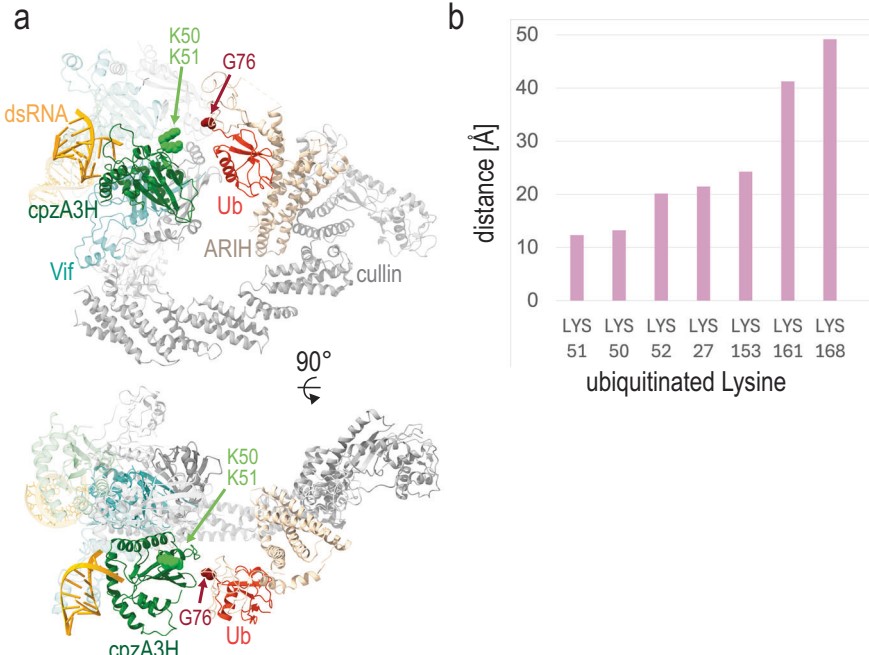

**Fig. 8 | Model of cpzA3H·Vif·CUL5 E3 ligase complex.** Model based on the cryo-EM structure from this study and components from PDB IDs 7B5L, 7B5M, 7ONI, 8FVJ[21,40,41]. **a** cpzA3H-VCBC dimer structure (this study) is modeled with the CUL5 E3 ligase machinery, with proteins color-coded as follows: cpzA3H in green, Vif in cyan, CBFβ in dark gray, EloBC in light gray, members of CUL5 E3 ligase complex in gray, ARIH in beige, and ubiquitin in dark red. Gly76 of ubiquitin and Lys50 and Lys51 of cpzA3H are shown as spheres. This model depicts transition state 2 that is about to transfer ubiquitin to the target protein. **b** Distances between the cpzA3H lysine side chains and Gly76 of the E3-bound ubiquitin. Ubiquitination of these lysines, mediated by the Vif-CUL5 E3 ligase machinery, was detected by mass spectrometry.

hydrophobic core at the primary interface between cpzA3H and Vif. This structural arrangement highlights the importance of Vif F39 in facilitating the degradation of A3H. On the other hand, Vif H48 is not positioned at the interface between A3H and Vif in our complex (this study) or the human A3H haplotype II VCBC complex[21] (NL4-3 Vif with N48H substitution was used for this study). Superimposition of these two A3H·Vif complex structures shows very similar main chain conformations (Vif residues 40–55) and comparable positioning of the H48 and N48 side chains (Supplementary Fig. 12). This region closely resembles the corresponding region in the A3G-VCBC structures (Supplementary Fig. 12), suggesting that this loop on Vif is structurally conserved and stable. In addition, H48/N48 is oriented toward CBFβ W73, and the structural conformation of CBFβ in relation to the Vif H48/N48 region is comparable across complexes with A3H, A3G, and A3F. Altogether, it remains elusive why H48 is important for degradation of human A3H haplotype II, despite accumulating structural evidence (our work and ref. 21) that this residue is not part of the interface between A3H and Vif.

The cpzA3H interaction interfaces appear to be highly conserved across HIV-1 and simian immunodeficiency virus (SIV) lineages. Notably, the Vif protein of SIV from Central African chimpanzees (*Pan troglodytes troglodytes*)—specifically the SIVcpzPtt strain MB897, which is phylogenetically closest to HIV-1 group M[51]—shares 78% amino acid identity and 92% similarity with HIV-1 (NL4-3) Vif. Eight residues critical for cpzA3H binding in HIV-1 Vif are conserved in SIVcpzPtt (MB897) Vif[14]. Consistent with this conservation, a previous study demonstrated that SIVcpzPtt (MB897) Vif efficiently antagonizes cpzA3H, similarly to HIV-1 Vif[35]. Furthermore, the structure of SIV Vif derived from red-capped mangabey was found to be highly similar to that of HIV-1 Vif, with a root mean square deviation (RMSD) of less than 1 Å[52], further supporting the notion of structural conservation across primate lentiviral Vif proteins. Collectively, these findings suggest that the structure and key residues of the natural antagonist, SIVcpz Vif, have

remained highly conserved among HIV-1/SIVcpz lineages to maintain effective A3H binding. This may be consistent with prior work proposing that an evolutionary equilibrium of HIV-1 Vif in human populations has yet to be reached[13].

Regarding the residues on cpzA3H that are modified by ubiquitination, our mass spectrometry showed that the primary ubiquitination sites are K50 and K51. This conclusion is supported by multiple lines of evidence, including our structure, mass spectrometry data, ubiquitination assays, and the model of the cpzA3H-Vif-CUL5 E3 ligase holoenzyme complex. Although the mass spectrometry data identified ubiquitination on sites beyond K50 and K51, i.e. K27, K52, K153 and K161, these ubiquitination events occurred with far lower frequency, and in the case of K52, only when K50 and K51 were already ubiquitinated. Furthermore, our cell-based ubiquitination assays demonstrate that only cpzA3H-7KR-50K and cpzA3H-7KR-51K recapitulate the degradation behavior of the WT cpzA3H, in agreement with their role as primary sites of ubiquitination critical for degradation. This is likely due to the local structure, as our model of the cpzA3H-Vif-CUL5 E3 ligase holoenzyme complex places K50 and K51 at 12 Å and 13 Å from the G76 of the ubiquitin (Fig. 8). All other lysines are further away, and this greater distance makes ubiquitin transfer less favorable. In addition to the local structural preference, the interaction between protomers may influence ubiquitination and obstruct the approach of ubiquitin. Our dimeric cpzA3H–VCBC structure is compatible with the binding of an extended CUL5 E3 ligase to each cpzA3H–VCBC protomer, as illustrated in Supplementary Fig. 13. The charged ubiquitin molecules (Ub_1 and Ub_2, shown in red) have unobstructed paths toward the Vif-proximal cpzA3H molecules, potentially enabling the formation of isopeptide bonds with target lysine residues. The other sites of ubiquitination we identified in mass spectrometry experiments (especially K153 and K161, and to lesser extent K27) may be reflective of our in vitro reaction conditions and may not be relevant in cells, although this remains to be dissected in future studies.

Importantly, our structure captures two A3H molecules bound to dsRNA. The dsRNA-mediated A3H dimer was proposed to be physiologically relevant, based on previous biochemical studies[15,34,53], structural characterization of the cpzA3H bound to dsRNA[33], and human and pig-tailed macaque A3H bound to dsRNA[31,32]. In agreement with the cpzA3H/dsRNA structure, we also observed that dsRNA makes specific contacts with the two A3H molecules in each protomer of the cpzA3H-VCBC complex. This positions the two A3H molecules in two distinct locations with respect to the rest of the complex. Whereas one of the A3H molecules is positioned close to Vif and oriented towards the central region of the cpzA3H·Vif·CUL5 E3 ligase holoenzyme complex (see Fig. 8), the other A3H molecule is positioned away from this region. The position of the proximal A3H molecule (A3H^P) agrees with the location of the A3H captured in the recent cryoEM structure of the human A3H-VCBC complex that visualized only one A3H molecule[21]. However, the position of distal A3H (A3H^D) has not been described before. Based on our structure, ubiquitination studies and the holoenzyme modeling, we propose that A3H^P is the main target for ubiquitination, with K50 and K51 being the primary targets for this modification. Given the location away from the incoming ubiquitin, we believe that A3H^D is not directly targeted by the ligase; however, we propose that its presence, together with the dsRNA, ensures the correct orientation and positioning of A3H^P, Vif, and the CUL5 components. Future studies will be needed to dissect this mechanism further and establish the role of A3H^D in this process.

In addition to mediating A3H dimerization, dsRNA binding plays a major structural role in the cpzA3H-VCBC complex. Not only does dsRNA serve as a platform for recruiting the A3H dimer to the complex, but it also seems essential for stabilizing the larger dimer formed by two cpzA3H-VCBC assemblies. We observed specific interactions between dsRNA of one protomer and Vif of the other protomer, as validated by site-selective Vif mutagenesis whereby mutations of residues involved in the Vif-dsRNA interface decrease the ability of Vif to degrade A3H in cells. As stated above, given that RNA has a profound impact on A3G-Vif binding and Vif-induced A3G degradation[18–20], we speculate that RNA binding might be an essential feature that directs A3s towards Vif-dependent proteasomal degradation. This is further supported by a recent preprint that reports that in the absence of dsRNA binding, A3H is transported into the nucleus and degraded in a process that does not involve Vif or CUL5, and depends on UBR4, UBR5, and HUWE1 as key ubiquitin E3 ligases[54]. This study suggests that the region on A3H that would normally be involved in dsRNA binding is recognized by these ligases in the absence of dsRNA binding. Therefore, dsRNA binding may function as a critical multipronged regulator of A3H, allowing it to recognize and neutralize the virus through hypermutations of the viral genome, enabling Vif to engage A3H while bound to the dsRNA and, when absent, allowing UBR4, UBR5, and HUWE1 to bind and prevent A3H from damaging host genome integrity, the activity that has been linked to cancer[54,55].

Collectively, our study advances our understanding of Vif-mediated A3H ubiquitination, highlights a more central role for dsRNA binding in this process, and determines molecular features that affect the efficacy of Vif-mediated A3H degradation. Given that this is one of the primary mechanisms HIV-1 uses to overcome host's resistance, our results have important implications for understanding this viral pathogen.

## Methods

### Protein expression and purification

For recombinant GST-cpzA3H protein production, the *E. coli* expression system was used as previously described with slight modifications[33]. Briefly, pET-41 GST-cpzA3H was transformed into Rosetta 2(DE3)pLysS competent cells (Merck Millipore) and grown in Luria-Bertani (LB) medium containing kanamycin (50 μg/ml) and chloramphenicol (34 μg/ml) at 37 °C to an optical density at 600 nm of ~0.6. After cooling down on ice to 20 °C, the bacterial cultures were induced with 10 mg/ml lactose monohydrate in the presence of 1 μM ZnSO4. After 18 h at 20 °C, cells were harvested by centrifugation at 7000 × g at 4 °C for 15 min and resuspended in lysis buffer A [50 mM Tris-HCl (pH 7.4), 1 M NaCl, 10%(v/v) glycerol, 5 mM 2-mercaptoethanol (2-ME), 1%(v/v) Triton-X100, 500 μg/mL RNase A (Qiagen)]. The lysate was sonicated, centrifuged at 28,620 × g for 30 min, and filtered through a 0.8-μm pore-size membrane. The supernatant was applied to a glutathione Sepharose 4 Fast Flow (FF) column (Cytiva), pre-equilibrated with the lysis buffer. The GST-tagged protein was eluted with elution buffer A [50 mM Tris-HCl (pH 7.4), 500 mM NaCl, 2%(v/v) glycerol, 1 mM 2-ME, 40 mM reduced L-glutathione], and treated with thrombin (33.3 U/ml) (Cytiva) and RNase A (1.67 mg/ml) at 20 °C for 16 h. The cpzA3H was further purified with size-exclusion chromatography (SEC). The protein was loaded onto a Hiload 26/600 Superdex-75 column (Cytiva) pre-equilibrated with gel filtration buffer A [50 mM Tris-HCl (pH 7.4), 150 mM NaCl, 5 mM 2-ME, 300 mM L-arginine HCl]. To remove the residual GST tag, the protein solution was loaded onto a Ni-Sepharose 6 Fast Flow (FF) (Cytiva) column in tandem with a glutathione Sepharose 4 FF column. The flow-through fraction containing the cpzA3H recombinant proteins was concentrated to 40 μM using an Amicon Ultra-0.5 (Merck Millipore).

Purification of Vif/HIS6-CBFß/EloB/EloC complex was performed as previously reported with slight modifications[37,56,57]. HIS6-tagged CBFβ/Vif and codon-optimized EloC/EloB were co-expressed from pET-Duet and pRSF-Duet plasmids, respectively. One Shot™ BL21 Star™ (DE3) Chemically Competent *E. coli* cells (Thermo Fisher Scientific) transformed with pET-Duet Vif/HIS6-CBFß and pRSF-Duet EloB/EloC were grown in LB medium containing ampicillin (75 μg/mL) and kanamycin (50 μg/mL) at 37 °C to an optical density at 600 nm of -0.6. After cooling down to less than 20 °C on ice, the bacterial cultures were induced with 0.5 mM Isopropyl β-D-1-thiogalactopyranoside (IPTG) in the presence of 10 μM ZnSO4 at 20 °C for 18 h. Cells were harvested by centrifugation at 7000 × g at 4 °C for 15 min and resuspended in lysis buffer B [50 mM Tris-HCl (pH 8.0), 500 mM NaCl, and 5 mM 2-ME]. The suspension was sonicated, after which one-third volume of 3X TritonX-100 lysis buffer [50 mM Tris-HCl (pH 8.0), 500 mM NaCl, 5 mM 2-ME and 3% TritonX-100] was added, and incubated for 30 min at 4 °C. Next, the sample was subjected to centrifugation at 23,000 × g for 30 min at 4 °C and filtered through a 0.8-μm pore-size membrane. The cleared soluble fraction was applied to a Ni-Sepharose 6 FF. The column was washed with 40 column volumes of wash buffer [50 mM Tris-HCl (pH 8.0), 500 mM NaCl, and 40 mM imidazole]. The VCBC complex was eluted with buffer B [50 mM Tris-HCl (pH 8.0), 500 mM NaCl, 300 mM imidazole, and 5 mM 2-ME]. The complex was further purified using a two-step method of Q cation-exchange chromatography (Cytiva) with IEX start buffer [25 mM Tris HCl (pH 8.0), 25 mM NaCl, and 5 mM 2-ME], IEX wash buffer [25 mM Tris HCl (pH 8.0), 200 mM NaCl, and 5 mM 2-ME] and IEX elution buffer [25 mM Tris HCl (pH 8.0), 500 mM NaCl, and 5 mM 2-ME]. The VCBC complex was further purified using a Hiload 26/600 Superdex 200 column equilibrated with gel filtration buffer B [25 mM Tris-HCl (pH 8.0), 500 mM NaCl, and 5 mM 2-ME] and concentrated to 40 μM using Amicon Ultra 15 mL 10 K (Merck Millipore).

Purification of HIS6-tagged CUL5/RBX2 was performed as previously reported with slight modifications[56,57]. HIS6-tagged CUL5/RBX2 and RBX2 were co-expressed from pRSF-Duet and pET plasmids, respectively. pET-RBX2 was used to ensure over-expression of RBX2. The two plasmids were co-transformed into One Shot BL21 Star™ (DE3) Chemically Competent *E. coli* and grown in LB medium containing Ampicillin (75 μg/mL) and Kanamycin (50 μg/mL) at 37 °C to an optical density at 600 nm of -0.6. After cooling down to less than 16 °C on ice, the bacterial cultures were induced with 0.5 mM IPTG in the presence of 10 μM ZnSO4 at 16 °C for 18 h. Cells were harvested by centrifugation at 7000 × g at 4 °C for 15 min and resuspended in lysis buffer B. Cells were sonicated for 1 h, after which one-third volume of 3X TritonX-100 lysis buffer was added. Solution was incubated for 30 min at 4 °C with mixing.

Next, the samples were subjected to centrifugation at 23,000 g for 30 min at 4 °C and filtered through a 0.8-μm pore-size membrane. The cleared soluble fraction was applied to a Ni-Sepharose 6 FF for affinity purification. The column was washed with 40 column volumes of wash buffer and protein complex was eluted using elution buffer B. The heterocomplex was further purified using a Hiload 26/600 Superdex 200 column equilibrated with gel filtration buffer B, and then concentrated to 40 μM by using Amicon Ultra 15 mL 30 K (Merck Millipore).

pET41-GST-ARIH2 plasmid was transformed into One Shot™ BL21 Star™ (DE3) Chemically Competent *E. coli*. Cells were grown in LB medium containing Kanamycin (50 μg/mL) at 37 °C to an optical density at 600 nm of -0.6. After cooling down to less than 16 °C on ice, the bacterial cultures were induced with 0.5 mM IPTG in the presence of 10 μM ZnSO4 at 16 °C for 18 h. Cells were harvested by centrifugation at 7000 × *g* at 4 °C for 15 min and resuspended in lysis buffer B. The suspensions were sonicated, after which one-third volume of 3X TritonX-100 lysis buffer was added and incubated for 30 min at 4 °C with mixing. Next, the samples were subjected to centrifugation at 23,000 × *g* for 30 min at 4 °C and then filtered through a 0.8-μm pore-size membrane. The cleared soluble fraction was applied to a glutathione Sepharose 4 FF column for affinity purification. The GST-ARIH2 protein fraction was obtained using elution buffer A, and treated with thrombin (33.3 U/ml) (Cytiva) at 20 °C for 16 h. The protein was loaded onto a Hiload 26/600 Superdex-200 column (Cytiva) pre-equilibrated with gel filtration buffer B. To remove the residual GST tag, the protein solution was loaded onto a Ni-Sepharose 6 FF column in tandem with a glutathione Sepharose 4 FF column. The flow-through fraction containing the ARIH2 recombinant proteins was concentrated to 40 μM using an Amicon Ultra-0.5.

pET-HIS6-tagged UbcH7 plasmid was transformed into One Shot BL21 Star (DE3) *E. coli*. The transformed bacteria were grown in LB medium containing Ampicillin (75 μg/mL) at 37 °C to an optical density at 600 nm of -0.6. After cooling down to less than 20 °C on ice, the bacterial cultures were induced with 0.5 mM IPTG 20 °C for 18 h. Cells were harvested by centrifugation at 7000 × *g* at 4 °C for 15 min and resuspended in lysis buffer B. The suspensions were sonicated, after which one-third volume of 3X TritonX-100 lysis buffer was added and incubated for 30 min at 4 °C with mixing. Next, the samples were subjected to centrifugation at 23,000 g for 30 min at 4 °C and then filtered through a 0.8-μm pore-size membrane. The cleared soluble fraction was applied to a Ni-Sepharose 6 FF for affinity purification. The column was washed with 40 column volumes of wash buffer. The His6-UbcH7 protein fraction was obtained using elution buffer B. The eluted His6- UbcH7 was further purified using Hiload 26/600 Superdex 75 column equilibrated with gel filtration buffer B, and then concentrated to 25 μM using Centriprep 10 K.

## Cryo-EM data acquisition

CpzA3H-VCBC complex was formed by mixing the VCBC complex and cpzA3H in 1:1 molar ratio and incubating on ice for 15 min. The protein complex was purified using HiLoad 16/600 Superdex 200 pg column (Cytiva) column in a buffer containing 25 mM HEPES (pH 7.5), 500 mM NaCl, 0.25% (v/v) glycerol, and 1 mM Tris(2-carboxyethyl)phosphine (TCEP). The peak fractions were pooled together and concentrated to ~28 uM.

The freshly purified cpzA3H-VCBC protein complex was cross-linked with 20 molar excess bis-sulfosuccinimidyl suberate (BS3, Thermo Fisher Scientific) in RT for 30 min. The reaction was quenched by adding 50 mM Tris (pH 8.0). The complex was further purified by Superdex 200 Increase 10/300 GL column (Cytiva) in a buffer containing 25 mM HEPES (pH 7.5), 200 mM NaCl, 0.25% glycerol, and 1 mM TCEP. 0.4 mL for each fraction was collected.

For each fraction without concentration, 3.5 μl aliquots of 0.2-0.35 mg/ml purified cpzA3H-VCBC complex were applied to Quantifoil Cu R0.6/1.0 holey carbon 300-mesh grids (Electron Microscopy

Sciences) glow discharged for 25 sec, 20 mA. Grids were blotted and vitrified in liquid ethane using Vitrobot Mark IV (Thermo Fisher Scientific) at 6 °C with 100% humidity.

Cryo-EM data of the cpzA3H-VCBC complex were collected using Talos Arctica G2 electron microscope (Thermo Fisher Scientific) equipped with K3 direct electron detector and energy filter operated at 200 kV in electron counting mode. Movies were collected at a nominal magnification of 100,000× in super resolution mode (0.405 Å/pixel). 50 frames per movie were acquired, with a total dose of 50 e⁻/Å² per and a total exposure time of approximately 2.5 s. A total of 44,160 movies were recorded by automated data acquisition with EPU (Thermo Fisher Scientific), with defocus values ranging from -2.2 to -0.9 μm.

## Cryo-EM data processing

The movies were imported into cryoSPARC 4.2.1 software package[58] and subjected to patch motion correction, binning to physical pixel size, and CTF estimation. Movies with CTF resolution worse than 5.5 Å and relative ice thickness >1.1 were removed, leaving 30,934 movies for further processing. Reference-free blob particle picking was performed to generate 2D templates for auto-picking. A total of 6,580,020 particles were picked initially, extracted, and down-sampled by a factor of 4. The iterative rounds of 2D and ab initio classifications were performed. 215,818 particles were re-extracted with full resolution and non-uniform refinement was performed, which yielded a - 4.0 Å resolution map of the C1 structure (Supplementary Fig. 4).

C1 map was used to generate 50 2D classes that were used as templates to pick particles again. After rounds of 2D and 3D classifications, 201, 593 particles were used in non-uniform refinement to reconstruct a 3.6 Å resolution map, with C2 symmetry applied.

To further improve the quality and interpretability of the calculated maps, we subjected them to post-processing with EMready[36]. All resolution evaluation was performed based on the gold-standard criterion of the FSC coefficient at 0.143[59].

## Model building and refinement

For cpzA3H-VCBC complex, atomic models derived from two copies of crystal structure of chimpanzee A3H (PDB ID: 5Z98) and two copies of VCBC in complex with N-terminal CUL5 (PDB ID: 4N9F) were docked into the cryo-EM map using UCSF Chimera[60]. The fragment corresponding to CUL5 was removed. The model was refined with the phenix.real_space_refine module in Phenix, with secondary structure restraints and geometry restraints[61]. The protein main chains and side chain rotameters were adjusted manually in COOT[62] to match the calculated map. Both atomic models were subjected to iterative cycles of real-space refinement in Phenix and manual adjustments in COOT. The final models were validated by tools available in Phenix (Table 1) All structural figures were generated using UCSF ChimeraX[63] or PyMOL[64].

## Vif-mediated degradation assay

We used pTR600 FLAG-cpzA3H WT or W90A plasmid as previously described[14,33]. We also used pcDNA-HVif WT, N48H, or LAI plasmids as previously described[3,14]. The cpzA3H 7KR or FLAG-cpzA3H WT-P2A-cpzA3H W90A gene was synthesized commercially (Fasmac and GenScript). For the pTR600 myc-cpzA3H WT or pTR600 cpzA3H WT (no tag) expression plasmid, the DNA fragment was amplified by PCR from pTR600 FLAG-cpzA3H WT plasmid DNA using the PrimeSTAR Max DNA Polymerase (Takara Bio); the primer sets used are listed in Supplementary Table 1. The myc-cpzA3H WT, cpzA3H WT (no tag), cpzA3H 7KR or FLAG-cpzA3H WT-P2A-cpzA3H W90A DNA fragments were inserted into pTR600 empty vector between the PstI and AvrII sites. For each Vif or cpzA3H mutant, substitution mutations were introduced by oligonucleotide-directed PCR using appropriate primer sets (Supplementary Table 1). For analysis of the Vif-mediated cpzA3H degradation, the cpzA3H expression plasmid (0.25 μg) and Vif expression plasmid (2.75 μg) were co-transfected into HEK293T cells in

**Table 1 | Cryo-EM data collection, refinement and validation statistics. C1 cpzA3H–VCBC (EMDB-47805), (PDB 9E9V) and C2 (EMDB-47752), (PDB 9E93)**

| Data collection and processing | | |
|---|---|---|
| Microscope | Talos Arctica | Talos Arctica |
| Detection camera | Gatan K3 | Gatan K3 |
| Voltage (kV) | 200 | 200 |
| Magnification | 100,000 | 100,000 |
| Electron exposure (e − / Å$^2$) | 45 | 45 |
| Defocus range (μm) | 0.8 – 2.2 | 0.8 – 2.2 |
| Pixel size (Å) | 0.81 | 0.81 |
| Symmetry imposed | C2 | C1 |
| Initial particle images (no.) | 10,231,372 | 6,580,020 |
| Final particle images (no.) | 201,593 | 215,818 |
| Map resolution (Å) | 3.63 | 4.0 |
| FSC threshold | 0.143 | 0.143 |
| Map resolution range (Å) | 2.97-8.0 | 3.5-7.7 |
| **Model building and refinement** | | |
| Initial model used (PDB code) | 4N9F, 5Z98 | 4N9F, 5Z98 |
| Model resolution (Å) | 4.0 | 4.55 |
| FSC threshold | 0.5 | 0.5 |
| Model resolution range (Å) | 3.0-6.0 | 3.4-6.6 |
| Map sharpening B factor (Å)$^a$ | -135.9 | -143.6 |
| B factors (Å$^2$) | | |
| Protein | 83.51 | 160.61 |
| Nucleotide | 84.91 | 107.42 |
| Ligand | 102.2 | 182.24 |
| *R.m.s. deviations* | | |
| Bond lengths (Å) | 0.002 | 0.001 |
| (outliers >4 sigma) | 0 | 0 |
| Bond angles (°) | 0.397 | 0.405 |
| (outliers >4 sigma) | 2 | 6 |
| *Validation* | | |
| MolProbity score | 1.29 | 1.34 |
| Clashscore | 5.46 | 4.31 |
| Rotamer outliers (%) | 0.0 | 0 |
| CaBLAM outliers (%) | 1.3 | 1.19 |
| Cβ outliers (%) | 0 | 0 |
| *Ramachandran plot* | | |
| Favored (%) | 98.52 | 97.35 |
| Allowed (%) | 1.48 | 2.65 |
| Outliers (%) | 0.0 | 0.0 |
| *Model composition* | | |
| Nonhydrogen atoms | 8,773 | 10,936 |
| Protein residues | 965 | 1241 |
| Nucleotide residues | 32 | 32 |
| Ligands | ZN: 4 | ZN: 4 |
| *Real-space correlation* | | |
| CCvolume | 0.75 | 0.6 |
| CCmask | 0.75 | 0.61 |

12-well plates using FuGENE HD (Promega). At 40 h post transfection, the cells were lysed using Laemmli buffer (Bio-Rad) in the presence of 2.5% (vol/vol) 2-ME. To analyze the protein expression levels, we fractionated HEK293T cell lysates by SDS-PAGE (12% acrylamide gel) and transferred them onto Immobilon-P membranes (Merck-Millipore). The membranes were first probed with the appropriate primary antibodies. The FLAG-cpzA3H was detected with an anti-DYKDDDDK (FLAG)-tagged mouse monoclonal antibody (mAb) (1:2,000; Fujifilm Wako Pure Chemical Co.; cat #012-22384). The myc-cpzA3H, cpzA3H (no tag), cpzA3H 7KR, FLAG-cpzA3H, and WT-P2A-cpzA3H W90A were detected with an anti-A3H rabbit polyclonal antibody (1:2,000; Novus Biologicals; cat #NBP1-91682). The HIV-1 Vif proteins were detected using a mouse anti-HIV-1 Vif monoclonal antibody (1:1000; Abcam; cat#ab66643), and β-tubulin, which was used as a loading control, was detected with a rabbit anti-β-tubulin polyclonal antibody (1:2000; Abcam; cat #ab6046). The immunoblotted membranes were subsequently incubated with horseradish peroxidase-conjugated secondary antibodies. A goat anti-mouse IgG antibody (1:20,000; Thermo Fisher Scientific) and a goat anti-rabbit IgG antibody (1:20,000; Thermo Fisher Scientific) were used for FLAG, A3H, Vif, and β-tubulin, respectively. The proteins were visualized by enhanced chemiluminescence using SuperSignal West Dura (Thermo Fisher Scientific) and quantified using an ImageQuant LAS 4000 (Cytiva).

### In vitro neddylation of CUL5/RBX2
NEDD8, and NEDD8 E1 NAE1-UBA3, UBE2F were purchased from R&D Systems. Purified HIS6-tagged CUL5/RBX2 was neddylated as follows. Reactions were assembled in neddylation buffer (25 mM Tris-HCl pH 7.4, 100 mM NaCl, 10 mM MgCl2, and 1 mM ATP) by the sequential addition of 0.15 μM NEDD8 E1 NAE1-UBA3, 2 μM UBE2F, 24 μM NEDD8, and 12 μM HIS6-tagged CUL5/RBX2. Reactions were initiated by the addition of HIS6-tagged CUL5/RBX2, incubated at 26 °C for 20 min. The neddylation reaction was quenched with the addition of Dithiothreitol (DTT) (10 mM) for 15 min on ice. Products were further purified by Hiload 26/600 Superdex 200 column equilibrated with gel filtration buffer B, and then concentrated to 40 μM by using Amicon Ultra 30 K.

### In vitro reconstituted ubiquitination assay
Recombinant proteins Ubiquitin (Ub), UBE1, and UbcH3(CDC34) were purchased from R&D Systems. In vitro *reconstituted* ubiquitination assays utilized a pulse-chase ubiquitination assay format[65,66]. Briefly, Ub, was thioester-linked to E2 Ubiquitin Conjugating Enzyme (UbcH3 or UbcH7) in a "pulse" reaction incubating 10 μM UbcH3 or UbcH7, 15 μM ubiquitin, and 300 nM UBE1 in 25 mM Tris-HCl, 100 mM NaCl, 2.5 mM MgCl$_2$, 1 mM ATP, pH 7.4 for 10 min at 26 °C. The pulse reaction was quenched for 5 min on ice with 50 mM EDTA, and each ubiquitin was chased from UbcH3 or UbcH7 to cpzA3H onto CUL5. Pulse-chase ubiquitination reaction mixtures were prepared by adding the E2-Ub thioester conjugate (1 μM or 2 μM final concentration) to 1 μM NEDD8 - CUL5-RBX2 with or without 1 μM ARIH2 pre-incubated with 1 μM VCBC WT or R41A mutant and 1 μM cpzA3H. Reactions were performed in 25 mM Tris-HCl, 100 mM NaCl, 50 mM EDTA, pH 7.4 at 26 °C. Aliquots were quenched at the indicated times by mixing with SDS sample buffer with 15 μM DTT or frozen immediately on dry ice, and stored at -80 °C until analysis. To detect the Ubiquitinated cpzA3H proteins, samples were separated by SDS-PAGE (12% acrylamide gel) and transferred onto Immobilon-P membranes (Merck-Millipore). The membranes were first probed with the anti-A3H rabbit antibodies (NBP1-91682, Novus Biologicals).

### Identification of ubiquitinated lysine using mass spectrometry
Sample preparation: 300 μL of the in vitro ubiquitination mixture, containing cpzA3H protein at a concentration of 12 μg/mL, was divided into four samples. Two of the samples were digested with Trypsin, while the other two were digested with Lys-N using the standard in-solution digestion procedure. One sample from each digestion was used for the enrichment of K-ε-GG peptides, following the protocol from the PTM-Scan Ubiquitin Remnant Motif (K-ε-GG) Kit (Cell Signaling). The digested peptide mixtures, or the peptides after enrichment, were concentrated and desalted using C18 Ziptips (Millipore). The desalted peptides were reconstituted in 15 μL of 0.1% formic acid, and 12 μL of each peptide sample was analyzed by a 60-min LC/MS/MS run.

Nanospray LC/MS/MS Analysis and Database Search: The LC/MS/MS analysis of tryptic peptides for each sample was performed sequentially using a Thermo Scientific Orbitrap Exploris 240 Mass Spectrometer coupled with a Thermo Dionex UltiMate 3000 RSLCnano System. Peptides from the trypsin digestion were first loaded onto a peptide trap cartridge at a flow rate of 5 μL/min. Trapped peptides were then eluted onto a reversed-phase Easy-Spray Column (PepMap RSLC, C18, 2 μM, 100 A, 75 μm × 250 mm, Thermo Scientific) using a linear gradient of acetonitrile (3–36%) in 0.1% formic acid over 60 min at a flow rate of 0.3 μL/min.

The eluted peptides were ionized and introduced into the mass spectrometer using a Nano Easy-Spray Ion Source (Thermo) with the following settings: spray voltage at 1.6 kV and capillary temperature at 275 °C. Raw data files were searched against the human protein sequence database and the chimpanzee A3H protein sequence database using Proteome Discoverer 2.5 software (Thermo, San Jose, CA) with the SEQUEST algorithm. Carbamidomethylation of cysteines (+57.021 Da) was set as a fixed modification, while oxidation (+15.995 Da) of methionine, deamidation ( + 0.984 Da) of asparagine and glutamine, and ubiquitination ( + 114.043 Da) of lysine were set as dynamic modifications. The minimum peptide length was set to five amino acids. Precursor mass tolerance was set to 15 ppm, and fragment mass tolerance was set to 0.05 Da. The maximum false discovery rate (FDR) for peptides was set at 0.05. The resulting Proteome Discoverer Report (Supplementary Data 1) contains all identified proteins, peptide sequences, peptide spectrum match counts (PSM#), and MS1 peak areas.

### Molecular dynamics

Molecular dynamic simulations were performed with the Desmond version 7.2 simulation package and the OPLS4 forcefield[67,68]. All systems were prepared identically. Missing sidechains and loops were modeled with PRIME and ionization state of sidechains were determined with PROPKA3[69]. The system was solvated with the TIP3P water model in a dodecahedron periodic boundary condition. Counter ions were added to a final concentration of 150 mM to neutralize the system and to mimic physiological conditions. The system was equilibrated in the isochoric (NVT) and isobaric (NPT) ensembles at 300 K for 12 ps with restraints on solute heavy atoms. One additional equilibration step was performed without restrains for 24 ps in the NPT ensemble. Production simulations were carried out in the NPT ensemble at a temperature of 300 K for a total of 1 μs.

### Sedimentation velocity analytical ultracentrifugation

Sedimentation velocity was carried out at 50,000 rpm (195,650 x *g* at 7.0 cm) and 10 °C on a Beckman Coulter ProteomeLab XL-I analytical ultracentrifuge and An50-Ti rotor following standard protocols[70]. Samples of cpzA3H, VCBC, Vif(H43A/W70A/H80A)CBC and the purified equimolar mixture of cpzA3H and VCBC in 500 mM NaCl, 25 mM HEPES pH 7.4, 1 mM TCEP, and 0.025% (v/v) glycerol in two channel centerpiece cells, and data were analyzed in SEDFIT[71] in terms of a continuous c(*s*) distribution of sedimenting species. Protein partial specific volumes were calculated in SEDNTERP[72], accounting for the RNA bound to cpzA3H. The buffer density and viscosity were measured experimentally at 20 °C on an Anton Paar DMA 5000 density meter and Anton Paar AMvN automated micro-viscometer, respectively. Values were corrected to 10 °C.

### Immunoprecipitation assay

To assess the formation of cpzA3H (WT/W90A) heterodimers, immunoprecipitation was performed with slight modifications from a previously described protocol[73]. Briefly, HEK293T cells were seeded in T25 flasks and cotransfected with plasmids encoding FLAG-tagged and myc-tagged cpzA3H. At 40 h post-transfection, cells were harvested and lysed on ice for 30 min in lysis buffer containing 150 mM NaCl, 1 mM EDTA, 1% Triton X-100, 50 mM Tris-HCl (pH 7.4), and 1 μg/mL

RNase A. The lysates were clarified by centrifugation at 20,630 × *g* for 10 min at 4 °C. FLAG-cpzA3H/myc-cpzA3H heterodimers were immunoprecipitated using anti-FLAG M2 magnetic beads (Merck), followed by washes with lysis buffer. Bound proteins were competitively eluted with 300 μg/mL 3× FLAG peptide (Merck). Immunoprecipitated samples were separated by SDS-PAGE using e-PAGEL 12.5% gels (Atto Co.) and transferred to Immobilon-P membranes (Merck-Millipore). Membranes were probed with primary antibodies: mouse monoclonal anti-DYKDDDDK (FLAG) antibody (1:2,000; Fujifilm Wako Pure Chemical Co.; cat. #012-22384), mouse monoclonal anti-myc (9B11) antibody (1:2,000; Cell Signaling Technology; cat. #2276S), and rabbit polyclonal anti-β-tubulin antibody (1:2,000; Abcam; cat. #ab6046) as a loading control.

### Reporting summary

Further information on research design is available in the Nature Portfolio Reporting Summary linked to this article.

## Data availability

Atomic coordinates for C1 cpzA3H–VCBC and C2 cpzA3H-VCBC structures were deposited in the Protein Data Bank under accession code 9E9V and 9E93, respectively. Cryo-EM density maps and half-maps for C1 cpzA3H–VCBC and C2 cpzA3H-VCBC were deposited in the Electron Microscopy Data Bank under accession code 47805 and 47752, respectively. Other structures used in this study include 4n9f, 5z98, 8h0i, 7b5l, 7b5m, 7oni, 8fvj, 8fvi, 8cx2, 8e40, and 6nil. Source data are provided with this paper.

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

## Acknowledgements

The authors would like to thank Drs. Wazo Myint, Joseph Caesar, and Takayuki Nagae for their advice and suggestions. The authors would like to acknowledge Dr. Sergey G. Tarasov and Ms. Marzena A. Dyba at the Biophysics Resources of CCR/NIH for providing instruments and technical support. We gratefully acknowledge the use of core facilities supported by the OIST Scientific Computing and Data Analysis Section, Scientific Imaging Section, and Instrumental Analysis Section. We also extend our gratitude to NCI Frederick for access to their cryo-EM facility and FRCE computing resources. This work was supported in part with grant from the U.S. National Institutes of Health R01AI150478 for CAS and HM and ZIA BC011627 to KJW. For KAS and HM, this project has been funded in part with federal funds from the National Cancer Institute, National Institutes of Health, under contract 75N91019D00024. This work was funded in part by the Intramural Research Program of the NIH, The National Institute of Diabetes and Digestive and Kidney Diseases for RG. This study was supported in part by Grants-in-Aid for Scientific Research from the Japan Society for the Promotion of Science (JSPS) KAKENHI (JP18K14685, JP20K07533, and JP23K06568 to KM and JP15H04740 and JP22H02882 to YI) and grants for Research Programs on HIV/AIDS from the Japan Agency for Medical Research and Development (AMED) (JP17fk0410304 and JP23fk0410058h0001 to YI). This research was supported in part by AMED under the U.S.-Japan research cooperation based on the Memorandum of Cooperation between AMED and NIH. KAS was supported in part by the NIH Office of Intramural Training and Education's Intramural AIDS Research Fellowship. MW was supported by direct funding from OIST. The content of this publication does not necessarily reflect the views or policies of the Department of Health and Human Services, nor does mention of trade names, commercial products, or organizations imply endorsement by the U.S. Government.

## Author contributions

K.A.S., K.M., Y.I., and H.M. conceived of the project. K.A.S. and K.M. prepared and characterized samples with support from VB. K.A.S. collected and analyzed TEM data and generated atomic models, while K.M. conducted all cell-based assays and in vitro ubiquitination assays. B.A.H. performed molecular dynamics simulations under KJW's supervision, and R.G. carried out AUC experiments and analyzed the data. T.H.C. and M.W. collected a subset of cryo-EM data. H.M. and Y.I. designed and supervised the study and coordinated collaborations, with H.M., C.A.S., and Y.I. securing funding. K.A.S. drafted the initial manuscript, and all authors contributed to manuscript writing and editing. K.A.S. and K.M. contributed equally to this work; YI and HM are co-corresponding authors.

## Funding

## Competing interests

The authors declare no competing interests.
