## [Transparent Peer Review file · Nature Communications]

HIV-1 Vif Mediates Ubiquitination of the Proximal Protomer in the APOBEC3H Dimer to Induce Degradation

Corresponding Author: Dr Hiroshi Matsuo

Version 0:

Reviewer comments:

Reviewer #1

(Remarks to the Author)

The manuscript by Skorupka et al presents a 3.8 Angstrom cryo-EM structure of the cpzA3H-VCBC complex, revealing that the complex forms a dimer. Their structural analysis indicates that dimerization is primarily mediated by RNA-Vif and Vif-Vif interactions. The functional significance of the interface residues was confirmed through cpzA3H degradation assay. Additionally, the authors captured cpzA3H in a dimerized state bound to dsRNA and found that only the proximal cpzA3H was ubiquitinated and degraded in a Vif-dependent manner, whereas the distal cpzA3H remained unmodified. Furthermore, they identified K50 and K51 of the proximal cpzA3H as the primary ubiquitination targets. This study follows a publication reporting the cryoEM structure of human A3H, haplotype II bound to HIV-1 Vif by Xiaojiang Chen and coworkers (PMID: 37640699).

In both structures there is good agreement between the primary interface between A3H and Vif. In contrast there are differences in the quaternary structures: the Chen structure reports a monomer of VCBC bound to A3H and dsRNA. Whereas the present manuscript documents a dimer of A3H bound to dsRNA that is recognized by VCBC.

The finding by Skorupka et al that Vif interacts with a dimer of A3H bound to dsRNA is significant, for the A3H dimer is thought to represent the form of A3H that is packaged into virions when Vif is not expressed. Therefore, this study is an important addition to the field as it suggests Vif recognizes an A3H dimer bound to dsRNA. Publication in Nature Communications is recommended after the following comments are addressed.

Major:

1-The authors miss the opportunity to compare their structure with that published by Chen and coworkers. A structural alignment of the cpzA3H-VCBC protomer with the hA3H-VCBC monomer should be provided, along with the calculated RMSD, to highlight structural differences and similarities.

2-Related, the authors may wish to state the salient similarities and differences in the approach for determining the different structures. For example, Chen and colleagues used a variant of human A3H-Hap II (K97Q) that renders it more susceptible to degradation by HIV-1 Vif by increases the strength of the protein-protein interaction. Likewise, Skorupka et al use A3H from chimpanzees which encodes this residue in the wild-type protein. In contrast, Chen and colleagues use NL43 Vif containing the N48H substitution, whereas Skorupka et al do not. Based on the prior Vif structures and the data shown in Supplemental Figure 9, it is puzzling to this reviewer why the N48H mutation facilitates interactions between Vif and A3H reported by Simon and colleagues and in vitro biochemical studies by Chen and Colleagues (PMID: 22013041, 37640699). Would the authors care to speculate on this in the discussion?

3-The authors may discuss why they are able to observe a dimeric A3H-Vif assembly in contrast to Chen and colleagues. Both studies employed chemical cross-linking to prepare samples for cryoEM, is it possible that Skorupka et al observe a dimer because cross-linking was carried out at high protein concentrations (that favor dimer formation) compared to Chen and colleagues?

4-Does the cross-linked complex in cryoEM reflect the wild-type dimer observed by SV-AUC? This could be tested by SV-AUC analyses of mutants predicted to shift the monomer to dimer equilibrium. For example, lesions that disrupt the Vif-Vif interface reduce formation of the dimeric and higher order complexes. Alternatively, the authors could compare SV-AUC profiles of wild-type and the cross-linked complex used for cryoEM, where would predict an enrichment in populations of

dimer and higher order species.

5-It is assumed but not tested that mutating A3H (W90A) does not affect the homodimer equilibrium in A3H molecules in the absence of Vif. The authors should IP wild-type tag/untagged A3H and compare it to an IP with the W90A mutant to determine if the mutation affects the dimer equilibrium in A3H.

Minor:

6-The presentation of the dimer structure in Figure 1 is confusing. The authors might consider stating the number of copies of A3H, Vif, CBFbeta and EloBC that they are able to model into the density in the structure solved without symmetry (C1) and the one solved with C2 symmetry imposed. A cartoon accompanying the maps presented in Fig 1 would be helpful. (The color code in the Figure Legend indicates the subunit coloring but not the stoichiometry.)

7-The authors should report the total buried surface areas for the primary interface (between Vif and the proximal copy of A3H) (Fig 2); the Across protomers RNA-Vif and A3H-Vif interaction (Fig 3); and the Vif dimer interface.

8-The electrostatic potential of the RNA binding pocket of Vif should be shown in the same view as Figure 4a, and Figure 4c should be removed to improve clarity.

9-The rotation in the Vif1-Vif2 interface should be explicitly indicated in Supplementary Figure 7.

10-The authors should include a panel illustrating the interaction of the $\alpha 1$ helix of Vif with RNA bound to A3G to provide better visualization of the result in lines 246-249.

11-Lines 143-144: The authors should reference the relevant literature.

12-Line 147: Consider replacing "distal A3H" with "dsRNA" since the C2 map does not show density for distal A3H.

14-Figure 2b: Additional views should be provided, as the side chain of L125 and the main chain of R124 are barely visible. Hydrogen bonds should also be indicated using dashed lines.

15-The map used in Supplementary Figure 6a (C1 or C2) should be specified.

16-Line 183 it is stated that L125 of cpzA3H interacts with Vif residues through hydrogen bonds. Please indicate if the backbone of L125 is making hydrogen bonds or revise this sentence since the side chain of leucine is not a H-bond donor or acceptor.

17-Line 398: "H48" should be corrected to "N48."

18-Line 400: Clarify that the NL4-3 strain with N48H was used for structural determination in Ref. 21.

19-Line 495: The concentration of RNase A used for RNA digestion (1.67 mg/ml) should be confirmed.

20-Line 569: "UBE2L3" should be replaced with "UbcH7" for consistency with the main text and Figure 6B. Or perhaps the yeast coenzymes were used (UbcH3, UbcH7) were used for the assay in Fig 6B. Please explicitly state whether yeast or human proteins were used, and if the former, indicate the sequence identity between yeast and chimps and human orthologues (in the methods).

21-Line 583: "0.4 μ l" should be corrected to "0.4 ml."

22-Line 598: The literature reference for the cryoSPARC software should be included.

23-Line 261-262: "dsRNA might be a general feature that directs A3s towards Vif-dependent proteasomal degradation" is incorrect: prior cryoEM studies show that a single stranded ssRNA molecule bridges Vif and A3G(dsRNA would not fit in the interface) (PMID: 37419875; 36598981; 36754086). Please modify the sentence appropriately.

Reviewer #2

(Remarks to the Author)

Reviewer #3

(Remarks to the Author)

(General comments)

The manuscript, "HIV-1 Vif Mediates Ubiquitination of the Proximal Protomer in the APOBEC3H Dimer to Induce Degradation," reports 3.6-4.0 Å resolution cryo-EM structures of chimpanzee APOBEC3H in complex with HIV-1 Vif, human

CBF- β , EloB, and EloC. The structure was validated through extensive functional studies. This work expands the collection of Vif-APOBEC3 complexes reported in recent years and advances our understanding of key interactions between HIV-1 Vif and host APOBEC3 proteins. The authors presented the complex structure both with and without imposed rotational symmetry, revealing that the previously observed RNA-mediated dimeric A3H is present in one protomer of the non-symmetrized complex structure. Their structures revealed a novel dimeric form of the A3H-Vif/CBF- β /EloB/EloC complex, which they demonstrated to exist in solution. Additionally, they identified new interaction interfaces, including Vif-dsRNA and Vif-Vif interactions. To further assess the functional relevance of these interfaces in A3H degradation, the authors generated a panel of Vif point mutants and conducted degradation assays, confirming the importance of these interactions in the degradation process. Through these assays, they identified new Vif residues affecting the degradation of A3H (R15, L81, K160, P161 in Vif-RNA interface and H42, Y44, P49, and W70 for Vif-Vif interface), which would be of interest to the field. Their functional data also pointed that Vif targets only the proximal A3H protomer within the dimer. Authors carefully corroborated their observations with previous mutational studies (refs 14 and 15) and the preceding cryo-EM structures.

(Specific points)

1. Line 81-82: "A3H, which is unique among the A3 proteins as the only family member that functions as a dimer." needs to be carefully re-written as some A3 members, including A3A, A3C, and A3G, have been demonstrated to function as dimer. Please refer to the relevant literature (such as PMID: 25914058; PMID: 28158858; PMID: 28928403) and revise accordingly to ensure accuracy.
2. For smoother reading in the introduction, authors should state the rationale for using chimpanzee A3H in this study instead of using human homolog, and mention that cpzA3H is degraded by HIV-1 Vif and it's more sensitive to HIV-1 Vif than human homolog (ref 14) to provide context for its selection.
3. Vif surface mapping displaying the binding sites for A3H, dsRNA and Vif from the other protomer should be shown.
4. Sequence alignment between human and chimpanzee A3H with Vif-binding residues highlighted would be informative.
5. Insights into cpzA3H binding by its natural antagonist, SIVcpz Vif, should be discussed, with a focus on the model structure and the conservation of A3H binding residues.
6. In the cryo-EM image processing workflow, multiple rounds of 2D classification after the first round of Ab initio reconstruction is somewhat unusual. Have you tried 3D classification or heterogeneous refinement to classify good particles here instead?

(Minor points)

1. Line 398-400: Please double-check that reference 21 also used Vif from NL4-3 strain, but they introduced N48H mutation (taken from LAI strain) to achieve higher binding to A3H. In line 400, "LAI Vif variant has N48" should be "LAI Vif variant has H48".
2. Line 288-291: An extended Cul5 RING ligase modeled on the present dimeric structure would inform the readers, especially what orientation the cullin arm extends relative to each other in this dimeric model.
3. Line 377-379: Please consider referencing Cul3-BTB domain-containing proteins as close examples of the multimerization of Cullin E3 ligase-substrate receptor complex observed in this study. The relevant references could be PMID: 38710693; PMID: 37450587.
4. Line 81: A3H (A3H) should be APOBEC3H (A3H).

Version 1:

Reviewer comments:

Reviewer #1

(Remarks to the Author)

The authors have addressed our questions and comments, the manuscript is now suitable for publication in Nature Communications.

Reviewer #2

(Remarks to the Author)

Reviewer #3

(Remarks to the Author)

The authors have thoroughly addressed my concerns and comments in the revised manuscript. I have no further issues, and I support its publication in Nature Communications.

We would like to thank the reviewers for the time and attention dedicated to helping us improve our manuscript. We answered all comments point-by-point in the following sections. Reviewers' comments are shown in italic font and highlighted in gray. Newly added text in the revised manuscript are highlighted yellow.

Reviewer #1

Major:

1-The authors miss the opportunity to compare their structure with that published by Chen and coworkers. A structural alignment of the cpzA3H-VCBC protomer with the hA3H-VCBC monomer should be provided, along with the calculated RMSD, to highlight structural differences and similarities.

Thank you for your valuable suggestion. In response, we have included a new figure (**Supplementary Figure 6**) to facilitate comparison with the structure reported by Chen and coworkers, and we added following sentence to the Result section lines 212-216: “Our structure of each cpzA3H–VCBC protomer closely resembles the recently published hA3H–VCBC structure by Ito, Chen, and colleagues, with an RMSD of less than 1 Å for each component protein (**Supplementary Figure 6**). Key A3H residues involved in Vif binding are conserved between the two structures, however, our structure reveals A3H–Vif interactions across protomers, as shown in **Supplementary Figure 6 panel c.**”

2-Related, the authors may wish to state the salient similarities and differences in the approach for determining the different structures. For example, Chen and colleagues used a variant of human A3H-Hap II (K97Q) that renders it more susceptible to degradation by HIV-1 Vif by increases the strength of the protein-protein interaction. Likewise, Skorupka et al use A3H from chimpanzees with encodes this residue in the wild-type protein. In contrast, Chen and colleagues use NL43 Vif containing the N48H substitution, whereas Skorupka et al do not. Based on the prior Vif structures and the data shown in Supplemental Figure 9, it is puzzling to this reviewer why the N48H mutations facilitates interactions between Vif and A3H reported by Simon and colleagues and in vitro biochemical studies by Chen and Colleagues (PMID: 22013041, 37640699). Would the authors care to speculate on this in the discussion?

This observation remains intriguing. A comparison of all available cryo-EM structures (**Supplementary Figure 12**) did not reveal a clear structural basis for the apparent functional difference between asparagine (N) and histidine (H) at residue 48 of Vif. Specifically, the aromatic rings of H48 and W73 are not positioned to form either perpendicular or stacked π - π interactions. Since neither residue participates in A3H interaction, and the Vif and CBF β structures are locally and molecularly similar, it seems reasonable not to speculate on why N48H enhances A3H interaction.

3-The authors may discuss why they are able to observe a dimeric A3H-Vif assembly in contrast to Chen and colleagues. Both studies employed chemical cross-linking to prepare samples for for cryoEM, is it possible that Skorupka et al observe a dimer because cross-linking was carried out at high protein concentrations (that favor dimer formation) compared to Chen and colleagues?

Thank you for this thoughtful suggestion. One possible explanation may lie in differences in sample preparation. Ito et al. reported that MBP-fused A3H was mixed with VCBC or VCBCR; however, they did not specify the concentration at which cross-linking occurred. Our analytical ultracentrifugation experiments indicate that VCBC exists in a dynamic equilibrium among monomeric, dimeric, and multimeric states, even at low concentrations (see Supplementary Figure 2). During our sample preparation, we consistently observe a population of dimeric complexes even without cross-linking. It is therefore likely that dimerization is a transient event, which we were able to capture during the cross-linking process.

4-Does the cross-linked complex in cryoEM reflect the wild-type dimer observed by SV-AUC? This could be tested by SV-AUC analyses of mutants predicted to shift the monomer to dimer equilibrium. For example, lesions that disrupt the Vif-Vif interface reduce formation of the dimeric and higher order complexes. Alternatively, the authors could compare SV-AUC profiles of wild-type and the cross-linked complex used for cryoEM, where would predict an enrichment in populations of dimer and higher order species.

Thank you for this important suggestion. Based on insights from our cryo-EM structure, we generated a Vif triple mutant (H43A/W70A/H80A), targeting residues located at the Vif-Vif interface. We then performed analytical ultracentrifugation (AUC) using a non-cross-linked VCBC sample. Our results show that these mutations significantly reduced the presence of dimeric and higher-order oligomeric forms (see updated **Supplementary Figure 2 panel e and f**). We have incorporated this new AUC result into the Results section lines 297-302. This observation supports the conclusion that the cryo-EM cross-linked sample captured a dimeric structure that also exists, albeit transiently, under non-cross-linked conditions.

5-It is assumed but not tested that mutating A3H (W90A) does not affect the homodimer equilibrium in A3H molecules in the absence of Vif. The authors should IP wild-type tag/untagged A3H and compare it to an IP with the W90A mutant to determine if the mutation affects the dimer equilibrium in A3H.

As suggested by the reviewer, we performed co-immunoprecipitation (Co-IP) analysis of FLAG-cpzA3H and myc-cpzA3H heterodimers using both wild-type (WT) cpzA3H and the W90A mutant. We added a new figure (**Supplementary Figure 9**) and text in the Results section lines 344-347 as shown below:

“We confirmed that WT and W90A cpzA3H proteins form heterodimers with comparable efficiency to WT-WT and W90A-W90A homodimers when FLAG- and myc-tagged cpzA3Hs were coexpressed in HEK293T cells”, and included the new data as

Supplementary Figure 9.

Minor:

6-The presentation of the dimer structure in Figure 1 is confusing. The authors might consider stating the number of copies of A3H, Vif, CBFbeta and EloBC that they are able to model into the density in the structure solved without symmetry (C1) and the one solved with C2 symmetry imposed. A cartoon accompanying the maps presented in Fig

1 would be helpful. (The color code in the Figure Legend indicates the subunit coloring but not the stoichiometry.)

Thank you for this helpful suggestion. In response, we have adjusted the color scheme to improve clarity and readability, as recommended. Additionally, we have revised the figure caption accordingly. Please see the **updated Figure 1**.

7-The authors should report the total buried surface areas for the primary interface (between Vif and the proximal copy of A3H) (Fig 2); the Across protomers RNA-Vif and A3H-Vif interaction (Fig 3); and the Vif dimer interface.

Thank you for this suggestion. We have now included all total buried surface areas between the relevant molecular interfaces in the appropriate sections of the manuscript. Specifically, the primary cpzA3H–Vif interface (692 Å²) is noted on line 159, the Vif–Vif interface (222 Å²) on line 276, the cpzA3H–Vif2 interface (245 Å²) on line 202, and the Vif–dsRNA interface (195 Å²) on line 246.

8-The electrostatic potential of the RNA binding pocket of Vif should be shown in the same view as Figure 4a, and Figure 4c should be removed to improve clarity.

Thank you for this helpful suggestion. We have revised **Figure 4** in accordance with the recommendation.

9-The rotation in the Vif1-Vif2 interface should be explicitly indicated in Supplementary Figure 7.

Thank you for this suggestion. We have revised the figure (**Supplementary Figure 8**) and its legend in accordance with the recommendation.

10-The authors should include a panel illustrating the interaction of the α 1 helix of Vif with RNA bound to A3G to provide better visualization of the result in lines 246-249.

Thank you for this suggestion. We have updated **Figure 4, panel b**, to show the α 1 helix amino acids involved in the interaction between Vif and A3G-bound RNA.

11-Lines 143-144: The authors should reference the relevant literature.

Thank you for this suggestion. The appropriate citations have been added accordingly, as outlined below.

(PDB ID: 4n9f) Guo Y, Dong L, Qiu X, Wang Y, Zhang B, Liu H, Yu Y, Zang Y, Yang M, Huang Z. Structural basis for hijacking CBF- β and CUL5 E3 ligase complex by HIV-1 Vif. *Nature*. 2014 Jan 9;505(7482):229-33. doi: 10.1038/nature12884. PMID: 24402281.

(PDB ID: 5z98): Matsuoka T, Nagae T, Ode H, Awazu H, Kurosawa T, Hamano A, Matsuoka K, Hachiya A, Imahashi M, Yokomaku Y, Watanabe N, Iwatani Y. Structural basis of chimpanzee APOBEC3H dimerization stabilized by double-stranded RNA.

Nucleic Acids Res. 2018 Nov 2;46(19):10368-10379. doi: 10.1093/nar/gky676. PMID: 30060196; PMCID: PMC6212771.

12-Line 147: Consider replacing "distal A3H" with "dsRNA" since the C2 map does not show density for distal A3H.

We have amended the text as suggested.

14-Figure 2b: Additional views should be provided, as the side chain of L125 and the main chain of R124 are barely visible. Hydrogen bonds should also be indicated using dashed lines.

We have revised **Figure 2** as recommended.

15-The map used in Supplementary Figure 6a (C1 or C2) should be specified.

Thank you for this suggestion. We have updated the figure caption accordingly. The revised caption now reads:

Supplementary Figure 7. - Structural Comparison of the Cryo-EM Model to the Crystal Structure of A3H.

Superimposition of the cpzA3H crystal structure (brown, PDB ID#5Z98) and cryo-EM cpzA3H structure from the C1 map of cpzA3H-VCBC complex (this study, cpzA3H:green, dsRNA: yellow), with (a) and without (b) the EM map.

16-Line 183 it is stated that L125 of cpzA3H interacts with Vif residues through hydrogen bonds. Please indicate if the backbone of L125 is making hydrogen bounds or revise this sentence since the side chain of leucine is not a H-bond donor or acceptor.

Thank you for pointing this out. We have amended the text in lines 185-187, and it now reads as follows:

“On the α 4 helix of cpzA3H^P, the side chains of E121 and R124 form hydrogen bonds with Y30 sidechain and K36 mainchain of Vif, respectively, while L125 engages in a hydrophobic interaction with F39 of Vif.”

17-Line 398: "H48" should be corrected to "N48."

Thank you for pointing this out. We have made the necessary change accordingly.

18-Line 400: Clarify that the NL4-3 strain with N48H was used for structural determination in Ref. 21.

We changed text accordingly in line 417 to clarify NL4-3 N48H Vif was used for the structural determination.

19-Line 495: The concentration of RNase A used for RNA digestion (1.67 mg/ml) should be confirmed.

We confirmed that the RNase A concentration was indeed 1.67 mg/ml for RNA digestion.

20-Line 569: "UBE2L3" should be replaced with "UbcH7" for consistency with the main text and Figure 6B. Or perhaps the yeast coenzymes were used (UbcH3, UbcH7) were used for the assay in Fig 6B. Please explicitly state whether yeast or human proteins were used, and if the former, indicate the sequence identity between yeast and chimps and human orthologues (in the methods).

As the reviewer suggested, we replaced UBE2L3 with UbcH7 as we used human UbcH7 in line 607.

21-Line 583: "0.4 μ l" should be corrected to "0.4 ml."

Thank you for pointing this out. We have made the change as recommended in line 621.

22-Line 598: The literature reference for the cryoSPARC software should be included.

Thank you for pointing this out. We have included appropriate reference shown below as reference #58:

Punjani, A., Rubinstein, J., Fleet, D. *et al.* cryoSPARC: algorithms for rapid unsupervised cryo-EM structure determination. *Nat Methods* 14, 290–296 (2017). <https://doi.org/10.1038/nmeth.4169>

23-Line 261-262: "dsRNA might be a general feature that directs A3s towards Vif-dependent proteasomal degradation" is incorrect: prior cryoEM studies show that a single stranded ssRNA molecule bridges Vif and A3G(dsRNA would not fit in the interface) (PMID: 37419875; 36598981; 36754086). Please modify the sentence appropriately.

Thank you for this suggestion. We have modified this sentence in lines 267-271 as shown below:

“Additionally, given that similar interactions have been documented in the APOBEC3G (A3G)–Vif structures^{18–20} and shown to be essential for Vif-induced A3G degradation²⁰, these results suggest that the presence of RNA may be a critical feature directing APOBEC3 proteins toward Vif-dependent proteasomal degradation.”

Reviewer #2 (Remarks to the Author)

(Specific points)

1. Line 81-82: “A3H, which is unique among the A3 proteins as the only family member that functions as a dimer.” needs to be carefully re-written as some A3 members, including A3A, A3C, and A3G, have been demonstrated to function as dimer. Please refer to the relevant literature (such as PMID: 25914058; PMID: 28158858; PMID: 28928403) and revise accordingly to ensure accuracy.

Thank you for this suggestion. We have revised the text accordingly in lines 81-83, as shown below:

“APOBEC3H (A3H)^{11,23-30}, which is unique among the APOBEC3 family members in that it forms a stable homodimer mediated by dsRNA, which bridges the protomer–protomer interface.”

2. For smoother reading in the introduction, authors should state the rationale for using chimpanzee A3H in this study instead of using human homolog, and mention that cpzA3H is degraded by HIV-1 Vif and it's more sensitive to HIV-1 Vif than human homolog (ref 14) to provide context for its selection.

Thank for this suggestion. We have added a sentence in the introduction to address this concern. More specifically, we added in lines 88-90:

“Previously, we and others reported that chimpanzee APOBEC3H (cpzA3H) is degraded by HIV-1 Vif more efficiently than human APOBEC3H derived from haplotype II^{14,35}, and that the cpzA3H protein exhibits relatively higher solubility³³.”

3. *Vif surface mapping displaying the binding sites for A3H, dsRNA and Vif from the other protomer should be shown.*

Thank you for the suggestion. We have generated a new **Supplementary Figure 11** illustrating the binding interfaces of A3H with dsRNA and Vif, and added a sentence in lines 384-386 as “To summarize Vif’s capacity to interact with multiple molecular partners, its binding interfaces are depicted in **Supplementary Figure 11.**”

4. *Sequence alignment between human and chimpanzee A3H with Vif-binding residues highlighted would be informative.*

Thank you for the suggestion. We have added a sequence alignment in **Supplementary Figure 6 panel c**, highlighting the residues involved in Vif binding.

5. *Insights into cpzA3H binding by its natural antagonist, SIVcpz Vif, should be discussed, with a focus on the model structure and the conservation of A3H binding residues.*

Thank you for this suggestion. We have added text to the Discussion section lines 429-443 to provide insights into the recognition of cpzA3H by its natural antagonist, SIVcpz Vif, as follows.

“The cpzA3H interaction interfaces appear to be highly conserved across HIV-1 and simian immunodeficiency virus (SIV) lineages. Notably, the Vif protein of SIV from Central African chimpanzees (*Pan troglodytes troglodytes*)—specifically the SIVcpzPtt strain MB897, which is phylogenetically closest to HIV-1 group M⁵⁰—shares 78% amino acid identity and 92% similarity with HIV-1 (NL4-3) Vif. Eight residues critical for cpzA3H binding in HIV-1 Vif are conserved in SIVcpzPtt (MB897) Vif¹⁴. Consistent with this conservation, a previous study demonstrated that SIVcpzPtt (MB897) Vif efficiently antagonizes cpzA3H, similarly to HIV-1 Vif³⁵. Furthermore, the structure of SIV Vif derived from red-capped mangabey was found to be highly similar to that of HIV-1 Vif, with a root mean square deviation (RMSD) of less than 1 Å⁵¹, further supporting the notion of structural conservation across primate lentiviral Vif proteins. Collectively, these findings suggest that the structure and key residues of the natural antagonist, SIVcpz Vif, have remained highly conserved among HIV-1/SIVcpz lineages to maintain effective A3H binding. This may be consistent with prior work proposing that an evolutionary equilibrium of HIV-1 Vif in human populations has yet to be reached¹³.”

6. *In the cryo-EM image processing workflow, multiple rounds of 2D classification after the first round of Ab initio reconstruction is somewhat unusual. Have you tried 3D classification or heterogeneous refinement to classify good particles here instead?*

Thank you for this comment. The 2D classification following *ab initio* reconstruction was used to reduce particles within the preferred orientation class, rather than to distinguish between discrete structural states. In more recent versions of cryoSPARC (v4.5 and later), we would instead utilize the 'rebalance 2D classes/orientations' function for this purpose. As part of our structure determination pipeline, we also employed both heterogeneous refinement and 3D classification. However, since these steps did not result in improvements to the final map used for model building, they were not included

in the final reported workflow.

(Minor points)

1. Line 398-400: Please double-check that reference 21 also used Vif from NL4-3 strain, but they introduced N48H mutation (taken from LAI strain) to achieve higher binding to A3H. In line 400, "LAI Vif variant has N48" should be "LAI Vif variant has H48".

Thank you for this comment. We added following text:

"NL4-3 Vif with N48H substitution was used for this study"

2. Line 288-291: An extended CUL5 RING ligase modeled on the present dimeric structure would inform the readers, especially what orientation the cullin arm extends relative to each other in this dimeric model.

Thank you for this suggestion. We have included **Supplementary Figure 13** to illustrate two CUL5 ligases modeled on the dimeric cpzA3H-VCBC structure, and added a sentence in the Discussion section lines 459-463 as shown below:

"Our dimeric cpzA3H-VCBC structure is compatible with the binding of an extended CUL5 E3 ligase to each cpzA3H-VCBC protomer, as illustrated in **Supplementary Figure 13**. The charged ubiquitin molecules (Ub_1 and Ub_2, shown in red) have unobstructed paths toward the Vif-proximal cpzA3H molecules, potentially enabling the formation of isopeptide bonds with target lysine residues."

3. Line 377-379: Please consider referencing Cul3-BTB domain-containing proteins as close examples of the multimerization of Cullin E3 ligase-substrate receptor complex observed in this study. The relevant references could be PMID: 38710693; PMID: 37450587.

Thank you for this suggestion. We added these references in line 399.

4. Line 81: A3H (A3H) should be APOBEC3H (A3H).

We have made change as suggested.